# Isocitrate Dehydrogenase Mutations in Myelodysplastic Syndromes and in Acute Myeloid Leukemias

**DOI:** 10.3390/cancers12092427

**Published:** 2020-08-26

**Authors:** Ugo Testa, Germana Castelli, Elvira Pelosi

**Affiliations:** Department of Oncology, Istituto Superiore di Sanità, Viale Regina Elena 299, 00161 Rome, Italy; germana.castelli@iss.it (G.C.); elvira.pelosi@iss.it (E.P.)

**Keywords:** leukemia, gene mutations, targeted therapy, isocitrate dehydrogenase

## Abstract

Acute myeloid leukemia (AML) is a heterogeneous disease generated by the acquisition of multiple genetic and epigenetic aberrations which impair the proliferation and differentiation of hematopoietic progenitors and precursors. In the last years, there has been a dramatic improvement in the understanding of the molecular alterations driving cellular signaling and biochemical changes determining the survival advantage, stimulation of proliferation, and impairment of cellular differentiation of leukemic cells. These molecular alterations influence clinical outcomes and provide potential targets for drug development. Among these alterations, an important role is played by two mutant enzymes of the citric acid cycle, isocitrate dehydrogenase (IDH), IDH1 and IDH2, occurring in about 20% of AMLs, which leads to the production of an oncogenic metabolite R-2-hydroxy-glutarate (R-2-HG); this causes a DNA hypermethylation and an inhibition of hematopoietic stem cell differentiation. *IDH* mutations differentially affect prognosis of AML patients following the location of the mutation and other co-occurring genomic abnormalities. Recently, the development of novel therapies based on the specific targeting of mutant *IDH* may contribute to new effective treatments of these patients. In this review, we will provide a detailed analysis of the biological, clinical, and therapeutic implications of IDH mutations.

## 1. Introduction

The human genome has five isocitrate dehydrogenase (*IDH*; EC 1.1.1.42) genes, coding for three distinct IDH enzymes, whose activities are dependent on either NADP (NADP^+^-dependent IDH1 and IDH2) or NAD (NAD^+^-dependent IDH3). Isocitrate dehydrogenase (*IDH*) genes encode the metabolic enzymes NADP^+^-dependent isocitrate dehydrogenase, involved in the catalyzation of the oxidative decarboxylation of isocitrate to synthesize an α-ketoglutarate (α-KG). There are two distinct IDH1 and IDH2 enzymes, showing a high degree of sequence similarity (about 70%), encoded by two distinct genes, *IDH1* located on 2q33 and *IDH2* located on 15q26.

The molecular structure of IDH1 enzyme was determined. This enzyme, localized in the cytoplasm, forms an asymmetric homodimer and exerts its biological activity through two active sites formed by both protein subunits. Each enzyme subunit is constituted by three different domains: A large domain (involving residues 1–103 and 286–414), a small domain (involving residues 104–136 and 186–285), and a clasp domain (involving residues 137 to 185 (Figure 1)). The active enzyme domain is represented by a deep cleft formed by the large and small domains of one subunit and a small domain of the other subunit; at the level of the active site, the NADP-binding site, and the isocitrate-metal ion binding site that are present. The shallow cleft is formed by the two domains of one subunit and is involved in the control of enzyme conformation. Finally, the two clasp domains of the two subunits interact to form a double layer of four-stranded anti-parallel beta-sheets connecting the two subunits and the two active sites [1]. A self-regulatory mechanism controls the activity of IDH1 through regulation of substrate binding: In the inactive enzyme, Asp^279^ occupies the isocitrate binding site and forms hydrogen bonds between Asp^279^ and Ser^94^ and enables Asp^279^ to chelate a metal ion, thus inducing the active enzyme conformation [1]. The enzymatic reaction catalyzed by IDH1 starts with the binding of substrate (isocitrate) to its binding site, favored by an open conformation of one subunit of IDH1 and the semi-open conformation of the other subunit; following isocitrate binding, the enzyme assumes a closed conformation, resulting in the formation of the catalytically active enzyme [2].

The structure of IDH2 is similar to that of IDH1; in addition, IDH2 contains a 39 amino acid mitochondrial targeting sequence at its NH2-terminus. The active site is formed by a hydrophilic cleft formed between the large and the small domain. The amino acid residues R140 and R172 stabilize the substrate-binding site [3]. The analysis of the three-dimensional structure of IDH2 showed the existence of a repulsion mechanism involving Lys^256^ and a lysine-rich cluster located on the opposing site of the active center, required to maintain the substrate-binding site in an open conformation [3]. In line with these findings, Lys^256^ acetylation induces a reduction of IDH2 enzymatic activity [3].

The structural organization of IDH3 is different from that of IDH1 and IDH2. In fact, IDH3 is a heterodimer α2βγ composed of the αβ and αγ heterodimers: αγ heterodimer can be allosterically activated by isocitrate and ADP, while the αβ heterodimer cannot be allosterically regulated by these activators and this is due to a different conformation of β and γ subunit at the level of the allosteric site [4].

Both IDH2 and IDH3 are localized in the mitochondrial matrix and participate in the citric acid cycle for energy production, whereas IDH1 is localized in the cytoplasm and peroxisomes. IDH1 and IDH2 enzymes function as homodimers, use NADP^+^ as electron acceptor, and require the binding of a divalent metal ion, usually Mn^2+^ or Mg^2+^. IDH3 catalyzes the third step of the citric acid cycle while converting NAD^+^ to NADH in the mitochondria. IDH enzymes catalyze the oxidative decarboxylation of isocitrate to produce α-ketoglurate (α-KG, also known as 2-oxoglutarate) and concomitantly produce NADPH from NADP^+^. IDH enzymes also catalyze the reductive carboxylation of α-KG to form isocitrate and concomitantly produce NADP^+^ from NADPH.

The enzymatic reaction catalyzed by IDHs implies a two-step process, where a first step involves oxidation of isocitrate to oxalosuccinate, followed by decarboxylation of the carboxyl group beta to the ketone, forming α-KG (Figure 2). The IDH1 and IDH2 enzymes are structurally organized as homodimers, while IDH3 is organized as a heterodimer, composed by two alpha, one beta, and one gamma subunit. The main biologic/physiologic function of IDH1/2 is related both to the biosynthesis of essential metabolites in the context of the tricarboxylic acid (TCA) cycle and in providing, together with the pentose phosphate pathway, one of the two essential cellular systems for the generation of NADPH. NADPH is required as a redox system for maintenance of redox homeostasis and reductive biosynthesis, with the existence of two separate cytosolic and mitochondrial pools supporting reductive potential in their respective locations [5]. Cytosolic NADPH is mainly regenerated via the oxidative pentose phosphate pathway and in the reactions catalyzed by IDH, malate enzyme (ME) and aldehyde dehydrogenase (ALDH). Particularly, for that concerning IDHs, the reductive carboxylation of α-KG to isocitrate by IDH2 consumes mitochondrial NADPH, with citrate/isocitrate transported to the cytoplasm, where these metabolites can be oxidized to produce cytosolic NADPH [5]. The reverse cycle can be used to produce mitochondrial NADPH [5].

In addition to TCA, glutamine-glutamate-α-KG metabolism represents an important step in the physiologic effects of IDH and is a critical determinant in IDH-mutant tumors. In this metabolic pathway, glutamine is converted to glutamate during the biosynthesis of nucleotides and asparagine in the cytoplasm or alternatively in the mitochondria by glutaminase. Then, glutamate dehydrogenase or transaminases are able to convert glutamate to α-KG: The choice of which pathway is used is influenced by some oncogenic pathways, cell proliferation, and metabolic condition [6]. The hypoxic microenvironment present in tumors stimulates glutamine flux into TCA metabolism and it becomes the predominant carbon source for glutaminolysis and reductive carboxylation pathways [7].

*IDH1* and *IDH2* genes are frequently mutated in some tumors and represent the metabolic genes most frequently mutated in human cancers [8,9]. Thus, *IDH1/2* genes are mutated in 50–80% of low-grade gliomas and secondary glioblastomas, about 20% of acute myeloid leukemia (AML), 50–60% of chondrosarcomas, about 10% of intra-hepatic cholangiocarcinoma and 10% of melanomas [8,9]. The frequency of *IDH1* and *IDH2* mutations are different in various tumor types: Thus, *IDH1* and *IDH2* are almost equally frequent in AML, while *IDH1* mutations are predominant in gliomas, chondrosarcomas, and cholangiocarcinomas [8,9].

## 2. IDH Mutations in Clonal Hematopoiesis

The natural history of AML development was related to the so-called age-related clonal hematopoiesis (ARCH) occurring by age 70 in about 10% of healthy individuals [10]. This condition reflects an aging-associated accumulation of somatic mutations at the level of hematopoietic stem cells; the majority of these mutations are neutral and do not confer growth advantage and no positive selection occurs on cells bearing these mutations. The introduction of ultra-deep sequencing techniques that are capable of detecting mutations in less than 0.5% of cells showed the presence of miniscule clones in more than 95% of older subjects, aged >65 years. Mutations in genes involved in epigenetic regulation (*DNMT3A, TET2, ASXL1, IDH1, IDH2*) are responsible for the development of the majority of mutation-driven ARCH [11]. The major negative consequence of ARCH is related to an increased risk of developing a blood neoplasia, evaluated in terms of a risk of transformation per year of 0.5–1%.

This sequence of progressive steps toward leukemia is driven by various intrinsic and extrinsic mechanisms: Stochastic events, related to aging-associated mutagenesis may initially generate mutations in ARCH; selective pressure on somatic variants dictated by aging or exogenous stress determine clonal outgrowth and ARCH formation; additional mutational events and selection of mutational events associated with a better fitness underline the leukemic progression [12]. Mutational events at the level of genes involved in RNA splicing (*SRFS2*, *serine/arginine-rich splicing factor 2*), DNA methylation (*DNMT3A (DNA(cytosine-5)-methyltransferase 3 A), TET2 (tet methycytosine dioxygenase 2), IDH 1-2 (isocitrate dehydrogenase 1 or 2*)), chromatin modification (*ASXL1*, *ASXL transcriptional regulator 1*) or the cohesion complex (*STAG2*, *stromal antigen 2*) are observed both in ARCH and in MDS; the gain of mutations at the level of genes encoding transcription factors (such as *RUNX1*, runt-related transcription factor 1 and *CEBPA*, CCAAT/enhancer binding protein) or signal transduction proteins (such as *FLT3*, fms-like tyrosine kinase 3, *c-kit*) leads to the development of AMLs secondary to MDSs [13]. Patients developing directly de novo AMLs have *RUNX1*, *CEBPA*, *FLT3* or *MLL (mixed lineage leukemia)* mutations, but not mutations associated with MDS [13]. 

Recent studies have directly assessed the potential risk conferred by ARCH to develop AML. Thus, Desai et al. have performed a deep sequencing analysis on peripheral blood DNA of 212 women who were healthy at study baseline but developed AML during follow-up and for comparison in a group of age-matched controls that did not develop AML. The most common identified mutations included *DNMT3A* (36.7% compared to 18.8% of controls), *TET2* (25% of cases compared to 5.5% of controls), *TP53* (11% of cases compared to 0% of controls), *IDH1+IDH2* (8% of cases compared to 0% of controls), *SF3B1* (5.9% of cases compared to 1.1% of controls), *JAK2* (5.3% of cases compared to 0.6% of controls), and *ASXL1* (3.2% of cases compared to 3.3% of controls) [12]. The presence of ARCH was associated with a clearly increased risk of developing AML: Particularly, mutations in *IDH1, IDH2, TP53* (*tumor protein 53*)*, DNMT3A, TET2,* and spliceosome genes increased the risk of developing AML; increased progression to AML was seen for those with >1 mutated gene by targeted sequencing (increased complexity) and 10% variant-allele fraction; interestingly, all patients with *TP53* or *IDH1/IDH2* mutations developed AML [12]. The median time of AML progression in the studied cohort was 9.6 years [12].

McKerrel et al. have explored the occurrence of clonal hematopoiesis in 4219 normal individuals of various ages using 15 hot spot mutations (at the level of *DMT3A, JAK2, NPM1, SRSF2, SF3B1, IDH1, IDH2, NRAS, KRAS, KIT, FLT3*) by ultra-deep sequencing and observed clonal hematopoiesis in 0.8% of individuals <60 years and 19.5% in those >90 years [14]. Interestingly, *IDH1* or *IDH2* mutations were detected only in individuals >75 years [14].

## 3. IDH Mutations in Myelodysplastic Syndromes

Myelodysplastic syndromes (MDS) include a heterogeneous group of myeloid neoplasms, which are characterized by common manifestations of bone marrow failure with abnormal cell morphology and a high tendency to acute myeloid leukemia (AML). 

Papaemmanuil et al. have reported the analysis of genetic alterations occurring in MDSs and reported *IDH1-IDH2* mutations in about 7% of cases, with *IDH2* mutations being more frequent (about 4.5%) than *IDH1* mutations (about 2.5%). *IDH2* mutations were particularly enriched in the RAEB subtype of MDS, were mutually exclusive with *TET2* and *SF3B1* mutations, and were frequently associated with *SRSF2* mutations [15].

Haferlach et al. performed a very large analysis of the genetic abnormalities observed in 944 MDS patients and reported *IDH2* mutations in about 4% of patients and *IDH1* mutations in about 3% of patients [16]. Particularly, 1.6% of patients displayed *IDH1*-*R132* mutations, 4.1% had *IDH2-R140* mutations, and 0.1% had *IDH2-R172* mutations [16]. *IDH1* and *IDH2* mutations were frequently co-expressed with *SRSF2* and *DNMT3A* mutations and were virtually mutually exclusive with *TET2* mutations [16]. Molenaar et al. explored *IDH1/IDH2* mutations in 868 low-risk and 536 high-risk MDS and observed: A higher frequency of *IDH1/IDH2* mutations in high-risk than in low-risk MDS, with a similar frequency of *IDH1* and *IDH2* mutations in low-risk MDS, but a higher proportion of *IDH2* than *IDH1* mutations in high-risk MDS; variant allelic frequencies indicated that *IDH2* mutations are more frequently ancestral than *IDH1* mutations; the presence of *IDH1/IDH2* mutations was associated with poor overall survival, particularly in low-risk MDS [17]. Di Nardo et al. have reported the analysis of 1042 MDS patients and reported *IDH-2* mutations in 5.7% of cases [18]. Particularly, 1.6% of patients displayed *IDH1-R132* mutations, 4.1% had *IDH2-R140* mutations, and 0.1% had *IDH2-R172* mutations [18]. *IDH1-IDH2*-mutant MDSs display some peculiar clinicopathologic features, compared with MDS patients *IDH1-IDH2*-WT: Lower absolute neutrophil counts, higher bone marrow blast percentage, and a trend for higher platelet cell counts [18]. The distribution of *IDH1-IDH2*-mutant MDSs in various risk categories was similar to that observed for MDS-WT; however, *IDH*-mutant MDSs displayed a different cytogenetic pattern, with 60% diploid karyotype, with isolated trisomy 8 in 10% of cases, and other intermediate cytogenetics in 23% of cases [18]. Moreover, at the level of the co-mutation pattern *IDH1-IDH2*-mutant MDSs displayed a peculiar pattern compared to *IDH*-WT MDSs, characterized by: Absence of *TP53* mutations (compared to 17% in *IDH*-WT MDSs); absence of *FLT3-ITD* or *FLT3-D835* mutation (compared to 2% in *IDH*-WT MDSs); lower frequency of *RUNX1* (13% vs. 40%), *TET2* (8% vs. 35%), and *ASXL1* mutation (21% vs. 44%) [18].

Two models of progression from MDS to sAML have been proposed: (a) A linear model based on bulk sequencing data suggests serial mutation accumulation during disease progression from unmutated HSCs, to clonal hematopoiesis, to MDS and finally to sAML; (b) a non-linear clonal evolution model based on the evidence that accumulation of mutations in stem cell compartments gives rise to a highly diverse subclonal architecture in MDS stem cells: Some of these subclones generate MDS, while other subclones act as pre-AML and then AML stem cells [19].

The dynamics of clonal evolution in myelodysplastic syndromes with *TET2* and *IDH1-IDH2* mutations evolving to secondary AMLs was explored showing that: (i) Apparently, *TET2* and *IDH* mutations did not have any significant impact on overall survival; (ii) TET2 but not *IDH1-IDH2* mutations were significantly associated with progression to s-AML; (iii) *TET2* and *IDH1-IDH2* mutations are among the genetic events that contribute to MDS initiation, whereas biallelic *TET2* mutations represent a subclone during the MDS phase, expanding in the s-AML phase; (iv) *TET2* and *IDH1-IDH2* mutations are maintained during transition to s-AML [20].

Makishima et al. have evaluated genotyping features observed in a very large data set of MDS samples with low-risk, high-risk, and secondary AMLs (sAML); this analysis allowed identifying genes preferentially mutated in high-risk vs. low-risk MDS (type-2 mutations) and in high-risk vs. sAML (type-1 mutations) [21]. Type-1 mutations were represented by *FLT3, PTPN11, WT1, IDH1, NPM1,* and *IDH2* mutations; type-2 mutations are represented by *GATA2, NRAS, KRAS, IDH2, RUNX1, STAG2,* and *ASXL1* [21]. These observations indicate an increase of the frequency of *IDH2* mutations in the progression from low-risk to high-risk MDS and then to AML; the frequency of *IDH1* mutations clearly increases in the progression from high-risk MDS to AML [21]. Importantly, MDS patients with type-1 mutations, including also *IDH1* and *IDH2* mutations, display a significant shorter time to progression to sAML compared to patients who had type-2 mutations but lacked type-1 mutations [21].

Some studies have explored the prognostic impact of *IDH* mutations in MDSs. Thus, Patnaik et al. have explored a group of 277 MDS patients and reported *IDH* mutations in 12% of these patients: About 9.4% displayed *IDH2* mutations (all *R140Q*) and 2.6% *IDH1* mutations (mostly *R132S*) [22]. *IDH* mutational frequency changed in the different MDS subtypes: 4% in refractory anemia with ring sideroblasts, 12% in refractory anemia with excess blasts (RAEB-1), and 23% in RAEB-2 [22]. All but one case with *IDH1* mutations displayed normal karyotype and 50% of *IDH2*-mutated MDS showed normal karyotype [22]. In multivariate analysis, *IDH1* mutations but not IDH2 mutations were associated with shortened leukemia-free survival [22]. Wang et al. confirmed these findings in a group of 97 MDSs showing that patients with *IDH1* mutations displayed shorter overall and progression-free survival, whereas *IDH2* mutations did not have impact on OS and PFS [23].

As above reported, *IDH2* mutations in MDSs are frequently overlapped with *SRSF2*. This finding suggested the existence of cooperating mechanisms between these two types of mutations in promoting leukemogenesis. In line with this hypothesis, Yoshimi et al. showed that co-expression of mutant *IDH2* and *SRSF2* in murine bone marrow cells resulted in lethal myelodysplasia with proliferative activity in vivo and enhanced self-renewal at an extent higher than with either mutation alone [24].

In conclusion, the study of *IDH* mutations in MDS show several relevant findings, supporting a pathogenic role: *IDH* mutations are present in MDS at a frequency lower than that observed in AML; the frequency of *IDH1/IDH2* mutations increases from lower-risk to higher-risk MDS, thus suggesting a role in clinical progression; in a fraction of MDS, *IDH2,* and *IDH1* mutations are involved in the ancestral neoplastic clone.

## 4. IDH Mutations in AML

In 2009, Mardis et al. reported frequent occurrence of *IDH1/2* gene mutations occurring in AMLs, a finding that was later confirmed by the Genome Atlas Research Network [25,26]. These studies and others have provided evidence that *IDH1/2* mutations occur in about 20% of AML patients, including 6–16% *IDH1* mutations and 8–19% *IDH2* mutations. *IDH*-mutated AMLs are characterized by a preferential occurrence in older patients, a preferential normal cytogenetic profile or other intermediate-risk cytogenetics, an increased percentage of leukemic blasts in the bone marrow and peripheral blood at diagnosis, a more frequent association with *NPM1* and *FLT3* mutations, a frequent association with *DNMT3A* mutation, and mutual exclusivity with *TET2* and *WT1* mutations [27,28,29,30].

Particularly, *IDH1* is most frequently mutated at the level of the Arg residue (R132), changing the substrate-binding arginine of the catalytic domain for R132H, R132C, R132L or R132S residues. *IDH1* and *IDH2* mutations are mutually exclusive, although in some occasional AML patients, concurrent mutations in both *IDH1* and *IDH2* are observed [31]. In an extensive meta-analysis, Patel et al. observed that the most frequent *IDH1*-mut co-mutations were *NPM1* (60.4%), *FLT3-ITD* (25.3%), and *CEBPA* (9%) [32]. *IDH1*-*R132H* and *IDH1*-*R132C* mutants exhibit a different distribution pattern among AML genotypes. *IDH1*-mutated AMLs showed in 62% of cases also a *NPM1* mutation, in 48% a *DNMT3A* mutation, 23% a *FLT3-ITD* mutation, 16% a *NRAS* mutation, and 12% a *SRSF2* mutation [33]. Interestingly, 89% of *IDH1-R132H* patients showed a *NPM1* mutation, while in only 33% of *IDH1-R132C* patients a *NPM1* mutation occurred [33]. *IDH1-R132H* was mutually exclusive for *RUNX1, SRSF2,* and *ASXL1*, whereas *IDH1-R132C* was frequently associated with *SRSF2* (21%), *RUNX1* (24%), and *ASXL1* (18%) [33] (Figure 3). According to these findings, it was proposed that *IDH1-R132H* shows a typical de novo AML pattern, while *IDH1-R132C* shows a more s-AML-like genetic pattern, suggesting a frequent evolution from an MDS condition [33].

*IDH2-R140* mutations were more frequent than *IDH2-R172*, representing about 80% of all *IDH2* mutations occurring in AML [33,35]. The *IDH2-R140* mutations frequently imply the substitution of arginine with glutamine (R140Q), whereas *IDH2*-*R172* mutations involve arginine replacement with lysine (R172K) [33,36]. While *IDH2-R140* mutations display a pattern of co-mutations similar to that observed for *IDH1*-mutant AMLs, *IDH2-R172* mutant AMLs have a very limited pattern of co-mutations and usually do not display *NPM1* co-mutations [37]. Furthermore, *IDH2-R172*-mutant AMLs have a poor response to standard chemotherapy treatments and have higher relapse rate [38]. Given this peculiar pattern of limited co-mutations, it is not surprising that *IDH2-R172*-mutant AMLs, but not *IDH1-R132* and *IDH2-R140* AMLs, were indicated as a distinct molecular subgroup in the context of genomic classification of AMLs proposed by Papaemmanuil et al. [39]. In fact, according to this study based on the genomic characterization of 1540 AML adult patients, while *IDH2-R172*-mutant AMLs are mainly clustered in a distinct subgroup exhibiting a co-mutation pattern limited to *DNMT3A* mutations, both *IDH1*-mutant and *IDH2-R140*-mutant AMLs are scattered in different AML subgroups, including *NPM1*-mutated, chromatin-spliceosome-mutated, and not-classified AMLs.

Other studies have shown remarkable differences between different *IDH2*-mutant AMLs at the level of genetic landscapes [34]. High white blood counts were rarely observed among *IDH2*-*172*-mutated patients (10%), compared to 48% and 55% in *IDH1*-R132 and *IDH2-R140*-mutated patients [34]. Cytogenetic alterations were observed in a limited proportion (about 26%) of *IDH*-mutant AMLs; however, the frequency of an aberrant karyotype was higher in *IDH2-R172*-mutated patients (42%) than in *IDH1-R132* (21%) and *IDH2-R140* (23%)-mutated AMLs [34]. Interestingly, *NPM1* mutations were absent among *IDH2-R172* patients, while they were frequent among *IDH1-R132* (63%) and *IDH1-R140* (50%) patients [34] (Figure 3). *FLT3* was mutated rarely in patients with *IDH2-R172* (5%), compared to those with mutations in *IDH1-R132* (28%) or *IDH2-R140* (31%) [34]. *SRSF2* was much more frequently mutated in *IDH2-R140*-mutated patients (43%) than in patients mutated in *IDH1-R132*-mutated AMLs, compared to 22% and 16% in *IDH2-R140*- and *IDH2-R172*-mutated AMLs, respectively [34] (Figure 3). AML patients with *IDH1-R132*, *IDH2-R140,* and *IDH2-R172* differ in their morphological and genetic patterns. This pathologic and genetic background translates into a more favorable prognosis for *IDH2*-R172-mutated AMLs, thus supporting their classification as a separate entity.

*IDH1* and *IDH2* are recurrently mutated in some AML subtypes (Table 1) mainly represented by NPM1-mutated, RUNX1-mutated AMLs, and AMLs bearing MLL-partial tandem duplication (MLL-PTD) and trisomy of chromosome 11 [36,40,41,42,43,44,45,46,47,48].

The study of minimally differentiated AMLs, classified as AML-M0 according to the FAB classification showed some interesting properties related to the presence of *IDH* mutations, observed in about 29% of these leukemias. In these AMLs, *IDH2-R172* mutations are more frequent than *IDH2-R140* mutations [49]. These AMLs were characterized also by frequent *RUNX1* mutations (about 24%) [49].

The incidence of AML increases dramatically with age, reaching its maximum in the age range comprised between 70–90 years. Thus, it is particularly important to define the genetic alterations occurring in older patients, representing the large majority of AML patients. The studies above mentioned are usually performed in patients <60–65 years. Silva et al. have explored the mutational profile of elderly AML patients and reported the occurrence of *IDH1* mutations (mostly associated with *DNMT3A* mutations) in about 17% of cases and of *IDH2* mutations (mostly associated with *DNMT3A* and *SRSF2* mutations) in about 11% of cases [50]. The ensemble of the mutational profile suggested a peculiar epigenetic landscape in older AML patients [50].

The comparative analysis of AML patients of <60 years, 60–74 years, and >75 years clearly showed that with aging there is an increase in the frequency of *TET2, ASXL1, RUNX1, IDH2,* and *TP53* mutation frequency [51]. In the older group of patients, *IDH1* mutations were observed in 9% of cases and *IDH2* mutations in 16–18% of cases [52]. None of the patients with *IDH1* mutations reached a complete remission following chemotherapy treatment, compared to a rate of 44% observed in the whole AML population [52]. This finding suggests that *IDH1* is a marker of chemorefractory disease and inferior prognosis in older AML patients [51]. In line with these findings, Heiblig et al. showed that in older AML patients *IDH2* mutations seem to confer a more favourable outcome compared to *IDH1* mutations (overall survival at three years 76% compared to 54%, respectively) [53]. Renaud et al. reported similar findings in a group of French AML patients older than 80 years, showing in these patients a 10% frequency of *IDH1* and 16% of *IDH2* mutations [54]. In adult AML patients, *IDH1* and *IDH2* mutations are usually mutually exclusive with *TET2* mutations; however, in older AML patients, it is more frequent to observe an *IDH/TET2* co-mutation pattern [54]. Standard treatments of older AML patients are based on conventional care regimens (reduced-intensity chemotherapy) or on hypomethylating agents, such as azacytidine. A recent study reported the response to the treatment of 485 older AML patients, 240 receiving azacytidine for seven days and 245 treated with conventional regimens (intensive chemotherapy, low-dose cytarabine or best supportive care only) [55]. In these patients, the most frequent gene mutations were *DNMT3A* (27%), *TET2* (25%), *IDH2* (23%, R140 15%, and R172 8%), *TP53* (21%), *IDH1*(9%); among *IDH*-mutant AMLs, 4/14 *IDH1*-mutant and 8/36 *IDH2*-mutant AMLs are associated with a poor-risk cytogenetics [55]. The median overall survival of *IDH2*-mutant AMLs was similar when treated with azacytidine (12.6 months) or with conventional regimens (12.5 months) [55]. A very recent study explored the frequency of the main driver gene mutations in a group of 325 Chinese AML patients of different ages, providing additional evidence that *IDH2, TP53, RUNX1,* and *SF3B1* mutations have significantly higher incidences in 60 years and older AML patients compared to those with <60 years [56].

Several reports indicate that the frequency of *IDH1* and *IDH2* mutations is lower in pediatric AML compared to adult AML. In an initial study, Andersson et al. explored *IDH* mutations in a population of 515 pediatric acute leukemia patients: *IDH1* and *IDH2* mutations were very rare in ALLs (1/288) and more frequent in AML (3.5%), with a higher frequency in AMLs with normal karyotype (9.8%). In this pediatric AML population, 3/5 *IDH2* mutant AMLs display *FLT3* mutations [57]. Damm et al. confirmed these findings and reported a frequency of 4% of *IDH1* and *IDH2* mutations in a group of 460 pediatric AMLs; in these patients, *IDH* mutations were associated with an intermediate patient age, FAB M1/M2 and *NPM1* mutations [58]. Valerio et al. have reported the analysis of the mutational profiling of 65 karyotype normal pediatric AMLs, focusing on the analysis of genes acting as epigenetic regulators; in this group of AMLs they observed a frequency of *IDH1* and *IDH2* mutations corresponding to 10.8% and mutations in *TET2* and *DNMT3A* were found with a frequency of 4.6% each. Interestingly, 4/7 of the *IDH*-mutant AMLs were associated with *FLT3-ITD* mutations [59]. An extensive characterization of the genetic abnormalities observed in 993 pediatric AML patients showed a frequency of *IDH1* mutations of 1% and of *IDH2* mutations of 3%. *IDH1* and *IDH2* mutations were significantly more frequent among older than younger pediatric AML patients [60]. It is important to note that the mutational profile of pediatric AMLs is very different, with some gene mutations, such as *NPM1, DNMT3A, TET2, IDH1, IDH2,* and *TP53* mutations being markedly less frequent in pediatric than in adult AMLs, whereas other gene mutations, such as *NRAS, KRAS, KIT, WT1,* and *GATA2* mutations being clearly more frequent in pediatric than in adult AMLs [60].

## 5. IDH Mutations in AMLs Secondary to MDS

Secondary AMLs (s-AMLs) are related to transformation events of an antecedent diagnosis of myelodysplastic syndrome or myeloproliferative neoplasms, whereas therapy-related AMLs (t-AMLs) are related to a late complication of previous exposures to leukemogenic therapies.

Progression towards AML occurs in about 30% of MDS patients. Fernandez-Mercado et al. have explored the mutation patterns in a group of 33 secondary AMLs with normal karyotype (24 secondary to MDS and nine to CMML) and showed that in s-AMLs developed following a previous MDS, 13% of cases displayed *IDH1* mutations and 8.7% *IDH2* mutations; in these patients, frequent were the *ASXL1* mutations (41.7%), whereas *NPM1* mutations occurred in 12.5% of cases [61]. Pellagatti et al. have investigated the changes in mutational profile in 41 patients undergoing progression from MDS to AML. The most frequently mutated genes in these patients were *ASXL1, TET2, SRSF2, U2AF1, RUNX1,* and *TP53*. *IDH1* and *IDH2* genes were mutated in 7% and 10% of cases, respectively and this frequency did not change after progression to AML [62]. Mutations of genes involved in splicing (*SFSR2, U2AF1, ZRSF2*), chromatin modification (*EZH2, ASXL1*), and DNA methylation (*IDH1, IDH2, DNMT3A*) were present in pre-progression and post-progression samples for almost all cases harboring these mutations, supporting their early occurrence during disease development; in contrast, mutations of genes involved in signal transduction (*NRAS, KRAS*) and in transcriptional regulation (*RUNX1, ETV6,PHF6*) were frequently found only in post-progression samples, thus supporting that they represent a late event during disease development [62]. Makishima et al. in their analysis on the dynamics of clonal evolution in MDS reported the analysis of the mutational profiling of 33 secondary AMLs developed from a preceding MDS and showed that mutations in seven genes, including *IDH1, IDH2, FLT3, PTPN11, WT1, NPM1,* and *NRAS* were significantly enriched in s-AML compared to high-risk MDSs; the presence of these mutations in high-risk MDSs reduce their time of progression to AML [21].

Lindsley et al. have performed a detailed analysis of genetic abnormalities observed in s-AMLs [63]. In this study, the authors explored 93 patients with s-AML (developed following an antecedent MDS in most of the cases or from an antecedent CMML) [63]. The s-AML group is heterogeneous at the mutational level in that a part of these patients (about 11%) displays *TP53* mutations, the majority (about 66%) secondary-type mutations and the rest (about 23%) in that part possess *de novo-like* AML-related mutations, such as *NPM1* mutations [63]. The secondary-type AMLs are characterized by the presence of frequent mutations of genes commonly altered in MDSs, such as *SRSF2, U2AF1, ASXL1, SF3B1, ZRSR, BCOR*, and *STAG2* mutations [63]. Globally, in s-AMLs, *IDH1* and *IDH2* mutations are observed at a frequency similar to that observed in *de novo*-AMLs; however, the majority of *IDH1*- and *IDH2*-mutated s-AMLs cluster within the subgroup characterized by the presence of secondary-type mutations; *IDH1* and *IDH2* mutations were absent in the subgroup of s-AMLs displaying *TP53* mutations and in part were present in the subgroup of s-AMLs characterized by a *de novo-like* AML mutational pattern [63]. The mutational pattern observed in t-AMLs is similar to that observed in s-AMLs, including AMLs displaying *IDH1* or *IDH2* mutations [63]. Interestingly, also *de novo*-AMLs can be subdivided into the three groups according to the mutational pattern and gene expression profile: The highest frequency (32%) of *IDH1/IDH2* mutations was observed more among secondary-type AMLs than among *de novo-like* AMLs (20%) and at the lowest level in the TP53-mutated subgroup (10%) [63].

## 6. IDH Mutations in AMLs Secondary to Myeloproliferative Neoplasms

According to the World Health Organization (WHO) classification, four variants of myeloproliferative neoplasms (MPNs), associated with JAK2, CALR, and MPL gene mutations were identified: Polycythemia vera (PV), essential thrombocytopenia (ET), primary myelofibrosis (PMF), and prefibrotic PMF (pre-PMF) [64].

Targeted deep sequencing studies carried out on large cohorts of patients have provided evidence about the occurrence of *IDH1/IDH2* mutations in only a small minority of patients: In PV, 3% of *IDH2* mutations and 0% of *IDH1* mutations [65]; in ET 1% of *IDH2* mutations and 0% of *IDH1* mutations [66]; in PMF 5% of *IDH2* mutations and 1% of *IDH1* mutations [66].

Several studies have explored the occurrence of *IDH* mutations during the blast-phase of myeloproliferative neoplasms. In an initial small study of blast/leukemic phase of preexisting JAK2- mutated MPN, five of 16 patients displayed *IDH* mutations: Three of these patients displayed an *R132C IDH1* mutation, whereas two exhibited an *R140Q IDH2* mutation [67]. Pardanani et al. screened 200 patients with either chronic or blast-phase MPN for *IDH* mutations and detected a total of nine patients bearing *IDH* mutations (five *IDH1* mutations and four *IDH2* mutations): The cumulative *IDH* mutational frequency was about 4% for patients in chronic phase and 21% for blast-phase MPN [68]. In a multi-institutional study, 1473 patients with MPNs (1422 in chronic phase and 51 in blast-phase); a total of 38 IDH mutations were detected (47.5% *IDH1-R132*, 50% *IDH2-R140,* and 2.5% *IDH2-R172*): Among patients in chronic phase the cumulative frequency of *IDH* mutations was 0.8% in ET, 1.9% in PV and 4.2% in PMF, and 21.6% among patients in blast-phase [69].

Rampal et al. performed a genomic analysis of 36 patients with MPN undergoing leukemic transformation and observed about 10% of *IDH1* mutations and about 30% of *IDH2* mutations; the majority of *IDH2*-mutant cases displayed a high variant allelic frequency [70].

In 2018, Lasho et al. reported a detailed targeted next-generation sequencing of 75 MPN patients in blast-phase: 52% post-PMF, 27% post-ET, and 21% post-PV; prior to leukemic transformation, disease progression to myelofibrosis was observed in 60% of patients with post-ET AMLs and in about 44% of those with post-PV AMLs [51]. Twelve percent *IDH1* and 7% *IDH2* mutations were observed in these AML patients, with a differential distribution according to the MPN driver mutation [51]. Venton et al. have reported the mutational spectrum observed in 73 post-MPN AMLs; 12.5% of these patients displayed *IDH1-IDH2* mutations [71]. In these post-MPN AML patients the frequency of *IDH1-IDH2* mutations was higher in post-ET and post-PMF AMLs than in post-PV AMLs [71]. Interestingly, in 71% of *IDH*-mutated post-MPN AMLs, *IDH* mutations are associated with *SRSF2* mutations; conversely, *SRSF2* mutations are associated in 55% of cases with *IDH* mutations [71].

The impact of mutational profile on the rapid bone marrow fibrosis observed in some PMF patients was explored: ARCH-associated mutations, such as *TET2, ASXL1,* and *DNMT3A* mutations, detectable at disease presentation were not associated with fibrotic progression; in contrast, mutations rarely associated with ARCH, such as *SRSF2, IDH1-IDH2, U2AF1, SF3A1,* and *EZH2* are connected to rapid fibrosis development and were not detectable in cases staying free from fibrosis [72].

It is of interest to note that studies of mutational profiling carried out in primary myelofibrosis have shown the occurrence of *SRSF2* mutation in 17% of cases; *SRSF2* mutations were frequently associated with *IDH* mutations: In *SRSF2*-mutated PMFs, 13% displayed *IDH1* mutations, compared to 1% in *SRSF2* wild-type PMFs; in *SRSF2*-mutated PMFs, 16% displayed *IDH2* mutations compared to 2% in *SRSF2* wild-type PMFs [73]. PMF patients with concurrent *JAK2* and *IDH1-IDH2* mutations have shorter leukemia-free survival [74], thus suggesting that co-mutations in *JAK2* and *IDH1-IDH2* could cooperate to promote MPN progression and transformation. Studies in transgenic mice provided evidence that JAK2^V617F^ and neomorphic *IDH1-IDH2* mutations cooperate in vivo to drive an aggressive myeloproliferative disease; the combined mutation MP disease was characterized by an expanded pool of pathological stem/progenitor cells [75]. Double-mutant MPNs were sensitive to IDH inhibitors and, particularly to the combined JAK2 and IDH2 inhibitor treatment [75].

In conclusion, the studies carried out on *IDH1/IDH2* mutations in MPN disorders have shown a low frequency of these mutations during the chronic phase, with a marked increase during the blastic transformation, suggesting an important pathogenic role for *IDH* mutants in leukemic transformation, as supported by *JAK2/IDH*-mutant mice studies.

## 7. IDH Mutations in Therapy-Related AML and MDS

About 10–20% of all newly diagnosed AML/MDS have a history of previous exposure to cytotoxic drugs or to radiation therapy, mostly for treatment of solid tumors or of hematological malignancies and various non-malignant conditions.

The origin of therapy-related myeloid neoplasms seems to be linked at cellular level to preceding events of clonal hematopoiesis: Sequencing data of tumor samples and peripheral blood mononuclear cells from a large set of cancer patients allowed identifying clonal hematopoiesis (CH) in 25% of these patients, with 4.5% harboring potential leukemia driver mutations (CH-PD) [76]. CH was positively associated with increased age, prior radiation therapy, and tobacco use; importantly, CH and CH-PD are associated with increased incidence of subsequent hematologic cancers [76].

Therapy-related myeloid neoplasms are a group of disorders comprising therapy-related AML (t-AML), MDS (t-MDS), and MDS/MPN (t-MDS/MPN), occurring as a late complication of cytotoxic therapy used in the treatments of a pathologic condition related to an oncological or not-oncological disease [77].

The study of 140 t-AML patients showed *IDH1/IDH2* mutations in 7% MDS and 12% of t-AML: Less frequently with *IDH1* (3/12 cases of *IDH*-mutated tumors) and more frequently with *IDH2* mutations (9/12 cases IDH-mutated) [78].

Lindsley et al. reported a detailed analysis of the mutational profiling of 101 t-AMLs: T-AMLs resulted in being a heterogeneous disease that, according to the genetic ontogenic-based classification, can be subdivided into three subgroups: 30% with a s-AML pattern; 23% with a *TP53*-mutated pattern; 47% with *de novo*-AML-mutated pattern. Seventeen percent of these t-AML patients displayed *IDH1/IDH2* mutations [63]. The distribution of *IDH1/IDH2* mutations was highly variable among these three subgroups: 33% of *IDH*-mutant among secondary-type t-AMLs; 8.5% of IDH-mutant among *TP53*-mutated t-AMLs; 10% of *IDH*-mutant among *de novo*/pan-AML t-AML [79]. T-AMLs with secondary-type mutations displayed an older age and had more recurrent driver mutations than t-AML with *de novo*/pan-AML mutations [63]. Voso et al. have explored the mutations of epigenetic regulators in 72 cases of t-MN (AML and MDS) and observed three *IDH1* and two *IDH2* mutations [80]. Young et al. have analyzed the mutational profiling of t-MDS and t-AML and have compared it to that of *de novo*-MDS and *de novo*-AML [81]. In t-MDS patients, no *IDH2* mutations were detected and about 3.5% of *IDH1* mutations; in t-AML, a frequency of *IDH1* and *IDH2* mutations were comparable to that observed in *de novo*-AML [81].

Singhal et al. have characterized the genetic abnormalities of 129 t-MN and observed that the mutational burden was similar in t-MN and in primary MDS (p-MDS); however, some notable differences exist in the mutational pattern between t-MN and p-MDS: (i) *TP53* mutations are more frequent in t-MN than in p-MDS; (ii) *SRSF2, SF3B1, U2AF1, CBL,* and *JAK2* mutations are less frequent in t-MN than in p-MDS [79]. *IDH1* and *IDH2* mutations are observed with a slightly higher frequency in t-MN than in p-MDS [79].

A recent study reported the molecular characterization of t-MN occurring after successful treatment of AML, a condition rarely observed [82]. In this study, 13 t-MDS and 12 t-AML were characterized. The mutations observed in these t-MN mostly affect epigenetic modifiers [82]. *IDH1* mutations were observed in 12% and *IDH2* in 16% of these t-MN. In the majority of cases, *IDH1* or *IDH2* mutations were observed both in originary AML and in t-MN or only in t-MN, but not in originary AML; only in one patient, an *IDH1* mutation was observed in the originary AML, t-MN, and was persistent at remission [82].

Kuzmanovic et al. have recently reported a very intriguing finding concerning *IDH* mutations in t-MN [83]. In a study focused on the analysis of genomics of therapy-related myeloid neoplasms, the genetic alterations observed in three groups of patients have been investigated: Primary MDS, t-MN, and a group defined as second MN (s-MN), related to patients who received only surgical treatment for a primary malignancy and developed myeloid neoplasms as a second cancer after surgical therapy of a primary tumor [83]. Very interestingly, *IDH1* mutations were 12 times more frequent in s-MN versus t-MN and three times more common versus p-MN [83]. These findings may support the view that these mutations predispose to develop a myeloid neoplasm and are less common in t-MN because they are suppressed by cytotoxic therapy [83]. To explain the origin of these t-MNs secondary to an initial AML it was proposed that the treatment of the originary AML eradicated the leukemic clones, with the exception of early clones of leukemogenesis responsible for clonal hematopoiesis; these clones acquire new gene mutations such as *IDH* mutations and emerge as a second leukemia neoplasm related to, but distinct from the originary AML.

## 8. Mechanisms of IDH-Induced Leukemogenesis

Most of leukemia-associated *IDH1* and *IDH2* mutations occur at the level of arginine residues present in the catalytic pocket of the enzyme, with the *IDH1* mutations occurring mostly at arginine 132 (R132H or R132C or R132L or R132S or R132G) and those of *IDH2* occurring mostly at arginine 172 or 140. These mutations reorganize the active site of the enzyme, causing an increased affinity for NAPDH to promote α-KG reduction at the expense of the principal substrate, isocitrate, and thus to confer to the mutant IDH1 or IDH2 protein a novel oncogenic enzymatic activity that is related to their capacity for allowing production of the R(-) enantiomer of the metabolite R-2-HG, which accumulates in *IDH*-mutant AMLs [84,85].

It was suggested that R-2-HG could represent the oncogenic mediator of *IDH* mutants in the leukemogenetic process. α-KG is a cofactor of many of the deoxygenases involved in the regulation of various key biologic processes, including nucleic acid repair, hypoxic response, chromatin modification and fatty acid metabolism, while 2-HG acts as an inhibitor of these deoxygenases [86,87]. Particularly, R-2-HG is a competitive inhibitor of histone demethylases and of the TET family of 5-methycytosine (5mC) demethylases [88]. Both R- and S-2-HG inhibit 2-oxyglutarate-dependent oxygenases with varying potencies: Thus, lysine demethylases were inhibited more efficiently than the hypoxia-inducible factor (HIF) prolyl hydroxylase [88].

TET2 catalyzes Fe(II)- and α-KG-dependent hydroxylation of 5-hydroxymethylcytosine (5hmC), involved in the regulation of gene expression. 5hmC triggers various mechanisms of DNA demethylation and inhibits the recruitment of methyl-DNA-binding transcriptional repressors to gene promoters. As above discussed, *IDH1-IDH2* and *TET2* mutations are mutually exclusive in myeloid neoplasia. Since R-2-HG inhibits TET2 enzymatic activity, it has been hypothesized that the effects of mutated *IDH* in AML are mainly due to R-2-HG-mediated TET2 inhibition and the consequent alterations to DNA methylation at the level of stem/progenitor myeloid cells [89]. This hypothesis is supported by the observations that expression of *IDH* mutants impaired TET2 catalytic function in cells and expression of either mutant *IDH1-IDH2* or *TET2* depletion impaired hematopoietic differentiation and increased stem/progenitor cell marker expression, suggesting a shared proleukemogenic effect [90,91]. In spite of these similarities, important clinical differences between *IDH*-mutant and *TET2*-mutant hematopoietic disorders and differences in alterations at the level of DNA damage repair mechanisms suggest that the oncogenic mechanisms of these mutated enzymes may differ [89]. This conclusion is supported by the observation that mutant *IDH1* induces a TET2- and DNA methylation-independent effect on the DNA damage response system in hematopoietic stem cells, resulting in a decrease of the number of these cells; these effects are due to histone modifications that lead to downregulation of the DNA damage sensor ATM [90]. Mutant IDH enzymes may promote tumor growth through mechanisms other than the reported inhibition of TET enzymes: Thus, mice expressing endogenous mutant *IDH1* display reduced numbers of hematopoietic stem cells, in contrast to *TET2*-deficient mice; mutant *IDH1* downregulates the DNA damage sensor ATM by altering histone methylation, determining impaired DNA repair, increased sensitivity to DNA damage, and reduced hematopoietic stem cell self-renewal, independent of TET2; ATM expression is decreased in primary *IDH1*-mutants AMLs [91]. According to these findings it was suggested that a model where a mutated *IDH1* displays two different oncogenic effects on hematopoietic stem cells induces an inhibition of DNA damage repair signaling and DNA repair at the level of long-term hematopoietic stem cells, through a mechanism dependent on ATM, but not on TET2; it also induces TET2-dependent alterations of DNA methylation driving the expansion of short-term hematopoietic stem cells and progenitor cells [91].

Thus, Wihle et al. have shown that myeloid leukemia cells overexpressing mutant *IDH* or that have been cultured in the presence of R-2-HG and *TET2*-mutated AML cells did not show similar methylation changes; the methylation patterns were compared to those observed in myeloid progenitor cells [92]. Through these studies the conclusion was reached that the differentiation state rather than the inhibition of *TET2*-mediated DNA demethylation is a major determinant of mutant IDH-associated hypermethylation observed in AML [92]. Studies carried out in the TF-1 *IDH2 R140Q* erythroleukemia model system showed that the *IDH2* mutant expression caused both histone and genomic DNA methylation changes that can be reversed when the *IDH2*-mutant activity is selectively inhibited: However, while histone hypermethylation is rapidly reversed within few days, reversal of DNA hypermethylation requires the course of weeks [93]. These changes in DNA methylation pattern could be related to induction of cell differentiation elicited by IDH inhibition [94].

Various experimental studies support a leukemogenetic role for R-2-HG. Thus, Losman et al. showed that the *IDH1 R132H* mutant promotes cytokine independence and blocks differentiation in hematopoietic cells; this effect is recapitulated by R-2-HG, but not by S-2-HG, despite the fact that S-2-HG more potently induces enzymes such as TET2 [95]. This paradox effect of the two 2-HG enantiomers is seemingly related to the ability of S-2-HG, but not R-2-HG, to inhibit EgIN prolyl hydroxylases [95].

Cell lines engineered to express mutant IDH proteins express markedly increased R-2-HG levels and impaired cellular differentiation [95,96].

Using two mouse models and a patient-derived xenotransplantation model, Chaturvedi et al. provided evidence that R-2-HG, but not S-2-HG and αKG, is an oncometabolite in vivo that does not require the mutant IDH1 protein to produce hyperleukocytosis and to accelerate the onset of murine and human leukemia [97]. Interestingly, the mutant IDH1 protein is a stronger oncogene than R-2-HG alone when comparable R-2-HG levels are achieved [97].

*IDH* mutants exert their pro-oncogenic effect by interfering with the differentiation program of hematopoietic cells. Thus, in 2000, Figueroa et al. analyzed the effects on stable expression of either an *IDH1* or *IDH2* mutant allele on hematopoietic cell differentiation in the 32D cultured mouse cells or in primary mouse bone marrow cells; in both of these cellular systems, the expression of an IDH-mutant enzyme induced an increase in stem cell markers and impaired myeloid cell differentiation [98]. Sasaki et al. reported the characterization of a conditional knock-in mouse model, in which the *IDH1-R132H* mutation was inserted into the murine *IDH1* locus and expressed in all hematopoietic cells or specifically in cells of the myeloid lineage [99]. These mutant mice displayed an increased number of early hematopoietic progenitors, impaired myeloid cell differentiation, anemia, splenomegaly and extramedullary hematopoiesis [99]. The hematopoietic cells of these animals displayed hypermethylated histones and changes to DNA methylation that were similar to those observed in *IDH*-mutant AMLs [99]. A third set of experiments provided evidence that enforced expression of an IDH-mutant enzyme in or exogenous administration of a soluble form of R-2-HG to the TF-1 human erythroleukemic cells promoted cytokine independence and blocked cell differentiation [1].

A fourth study was based on the development of leukemia through transformation of murine hematopoietic stem/progenitor cells with *IDH2* mutants in cooperation with FLT3-ITD or NRAS mutant alleles [100]. *NRAS* mutations were observed in 15% of *IDH*-mutant AMLs: *NRAS* mutations were equally observed among *IDH1 R132* and *IDH2 R140*-mutated AMLs, while were absent in *IDH2 R172*-mutated AMLs [34]. *FLT3-ITD* mutations are observed as co-mutations in 21% of IDH-mutant AMLs [34]; compared with *FLT3-ITD^+^/IDH-WT* patients, *FLT3-ITD/IDH^+^* double mutated patients had higher white blood cell counts and increased blast percentages at presentation; the frequency of *NPM1* mutations was significantly higher in the *FLT3-ITD^+^/IDH^+^* cohort, whereas *DNMT3A* mutations were lower [101]. The response to standard chemotherapy was comparable in *FLT3-ITD^+^/IDH^+^* and *FLT3-ITD^+^/IDH-WT* cohorts and there was no significant difference between *IDH1* and *IDH2*-mutated patients [101]. In a more recent study, Cortes et al. showed that *FLT3/IDH2* patients have better event-free survival and overall survival than *FLT3/IDH1* patients in frontline and relapsed/refractory setting when patients were treated with a FLT3 inhibitor in combination with cytotoxic chemotherapy or with low-intensity therapy (hypomethylating agents and low-dose cytarabine) [102].

Other studies have supported a role of Meis1 and HoxA9 in cooperation with *IDH1* or *IDH2* mutant to drive leukemia development in mouse models. Mutant *IDH1* alone was unable to transform hematopoietic cells, but consistently accelerated leukemia development induced by HoxA9 [103]. Kats et al. developed a mouse transgenic model of *IDH2-R140Q* mutation that has the capacity to be both tissue-specific and on/off inducible; using this genetic model, it was demonstrated that expression of the transgene elicited an on/off inducible 2-HG production that was comparable to that observed in AML patients [104]. Expression of mutant *IDH2* resulted in alterations within the hematopoietic compartment, characterized by an expansion of HSCs and a partial blockade of hematopoietic cell differentiation [104]. Development of compound transgenic models, in which the expression of mutant *IDH2* was combined with *Meis1* and *HoxA9*, led to the development of leukemic cells that were dependent on the expression/function of mutant *IDH* for their growth/survival; on the other hand, compound transgenic *IDH2-R140Q*; Flt mice showed that mutant *IDH* cooperates with *FLT3-ITD* in leukemia inhibition in vivo [104]. Ogawara et al. developed a peculiar model of IDH-dependent leukemia, in which mice were transplanted with NPM1+/− hematopoietic stem/progenitor cells co-transduced with four mutant genes (*NPMc, IDH2-R140Q, DNMT3A-R882H*, and *FLT3-ITD*) [105]. The resultant leukemias that developed in these animals were dependent upon the expression of mutant *IDH*, as supported by the observation that conditional deletion of *IDH2-R140Q* blocked 2-HG production and maintenance of leukemic stem cells, resulting in survival of the AML mice [105].

Gene expression studies carried out on leukemic cells bearing mutant IDH enzymes have clearly shown that *IDH1-R132H* mutation primes leukemic blasts to granulo-monocytic differentiation (as directly supported by the finding of an enrichment of key transcriptional factors regulating myelopoiesis, such as CEBPα, PU.1, RUNX1, CEBPβ, CEBPε) [106]. Particularly, analysis at the level of the *CEBPα*gene showed that *IDH1*-mutant AML cells have an increased occupancy of the promoter of this gene by H3K4me3, which is associated with expression of *CEBPα* and of its target genes [106]. Furthermore, the gene expression analysis also showed that the *IDH1-R132H* gene signature is particularly enriched in genes that are responsive to treatment with retinoic acid receptor (RAR) ligands, such as all-trans retinoic acid (ATRA) [106]. Importantly, in vitro treatment of *IDH*-mutant AML cells with ATRA resulted in induction of granulocytic differentiation, associated with a reduction in cell viability that occurred through induction of apoptosis [107]. In vivo ATRA treatment of immunodeficient mice grafted with human *IDH*-mutant AML cells resulted in a clear reduction of tumor burden [106].

Mugoni et al. have developed an unique mutant *IDH2* mouse model that evolved from an initial IDH2 dependence to an IDH2-independent status [108]. This model was used to understand some molecular changes associated with evolution towards IDH2 independence and to identify some vulnerabilities of IDH2-independent AML, showing that: (i) An increase in reactive oxygen species (ROS) may cause a genotoxic effect; (ii) enrichment of the tretinoin/retinoic acid pathway; (iii) suppression of LSD1 demethylase; (iv) upregulation of Pin1 prolyl isomerase [108]. These features suggest that *IDH2*-mutant AML cells are sensitive to the differentiation inducing activity of ATRA (all-trans retinoic acid) and to the proapoptotic effect of arsenic trioxide (ATO). This hypothesis was supported by experimental studies showing that the ATRA+ATO treatment is synergistic in its anti-tumor effects in a number of mouse and human mutant *IDH1/IDH1* leukemic models [108].

Other studies were focused on the identification of some metabolic/biochemical abnormalities in mutant *IDH* AML cells, representing vulnerabilities exploitable at the therapeutic level. Chan et al., through a large-scale RNA interference screen, have identified the anti-apoptotic gene *BCL-2* as synthetic lethal for *IDH1*-mutant AML cells [109]. *IDH1*- and *IDH2*-mutant primary human primary human AML cells were more sensitive than *IDH1/IDH2-WT* AML cells to ABT-199, a specific BCL-2 inhibitor [110]; this sensitization of mutant *IDH* AML cells is induced by R-2-HG-mediated inhibition of the activity of cytochrome c oxidase in the mitochondrial electron transport chain [110].

A very recent study provided evidence about a possible implication of long-non coding RNAs (lncRNAs) in the pathogenesis of differentiation block induced by *IDH* mutants in leukemia [111]. The expression of lncRNA Cancer Susceptibility CASC15 is inversely correlated with myeloid differentiation and is overexpressed in AMLs bearing *IDH* and *TET2* mutations [111]. CASC15 expression is higher in AMLs bearing *IDH* mutations without concomitant *DNMT3A* mutations compared to that observed in AMLs bearing both mutations [112]. *CASC15* expression was higher in *IDH2-R140*-mutant AMLs, compared to *IDH1-R132*- and *IDH2-R172*-mutant AMLs [111]. Introduction of mutant *IDH* in experimental models induced a marked enhancement of lncRNA *CASC15* [111].

In spite of the antagonism between *IDH* and *DNMT3A* mutations concerning the DNA methylation effects, these two epigenetic mutations were recently shown to cooperate to induce leukemia [113]. Leukemia-initiating cells isolated from a *DNMT3A* deficient mouse that expresses an *IDH2* mutant displays a megakaryocyte-erythroid progenitor-like phenotype, activates a stem-like gene signature, represses differentiated progenitor genes, and displays an epigenomic dysregulation [114]. Furthermore, targeted metabolomic profiling showed the overproduction of prostaglandin E2 in leukemic stem cells [113]. Stem/progenitor cells bearing both *DNMT3A* and *IDH* mutations are induced to differentiate by inhibitors of prostaglandin synthesis and by inhibitors of histone deacetylase [113].

## 9. IDH Mutations and DNA Methylation

*IDH1/2* mutations, together with *DNMT3A* and *TET2* gene mutations, contribute to an overall occurrence in AMLs of >40% of the mutations in genes involved in the regulation of methylation of genomic DNA [35]. Figueroa et al. showed that *IDH1-IDH* mutations, induced similar epigenetic alterations as *TET2* mutants [98]. Introduction of mutant *IDH* alleles into recipient cells induced a global DNA hypermethylation and impaired TET2 catalytic function [98]. Subsequent studies have shown that the hypermethylation defect observed in mutant *IDH* AMLs is related to the overproduction of the oncometabolite 2-hydroxyglutarate (2HG), responsible for inhibition of histone demethylases, with consequent DNA hypermethylation and block of cell differentiation [99]. A more detailed analysis of the DNA hypermethylation pattern observed in AMLs showed a widespread hypermethylation condition, preferentially targeting promoter regions and CpG islands neighboring the transcription start sites of genes [115].

These findings were corroborated through the analysis of mouse models based on the expression of *IDH* mutants into recipient normal hematopoietic cells. Thus, Sasaki et al. provided evidence that the expression of *IDH1*-*R132H* mutant induces a pattern of hypermethylated histones and changes to DNA methylation similar to those observed in human *IDH1*- or *IDH2*-mutant AMLs [116]. Furthermore, aberrant methylation was not restricted to promoter regions but, instead, targets also a significant proportion of intergenic and intronic regions [116].

A comprehensive DNA methylation profiling analysis of adult AMLs showed that differential methylation of non-promoter regulatory elements is a driver of epigenetic identity; through the enhanced reduced representation bisulfite sequencing (ERRBS) *IDH1-IDH2* mutant cases can be subdivided into a predominantly *IDH1*-mutant cluster (cluster 1) in which all mutant cases also harbored *DNMT3A* mutations, and a second cluster exclusively carrying *IDH* mutations without co-occurring *DNMT3A* mutations, almost of which were *IDH2* mutations [117]. Interestingly, mutations in *IDH* and *DNMT3A* had opposing and mutually exclusive effects on the epigenome; as a consequence, co-occurrence of both mutations resulted in epigenetic antagonism, with most CpG affected by either mutation alone, no longer affected by double-mutant AMLs [117].

Gebhard et al. have reported the profiling analysis of aberrant DNA methylation in AMLs, distinguishing the CG regions into those non-targeted by Polycomb (non-PcG) and those targeted by Polycomb (PcG); in the non-PcG, 10 methylation clusters were identified, with cluster 5 enriched in *IDH2* mutations and cluster 9 in *IDH1* mutations [118].

Recently, Vosberg et al. reported a detailed DNA methylation profiling study in AML patients, focusing on the analysis of the effects of mutations in commonly mutated genes (such as *NPM1, FLT3-ITD, DNMT3A, IDH1, IDH2, TET2,* and *WT1*) on DNA methylation profiles and differential gene expression [119]. Mutations at the level of *DNMT3A* resulted in global DNA hypermethylation, while alterations of *IDH1, IDH2, TET2,* or *WT1* resulted in global hypermethylation; mutations in *IDH1, IDH2, TET2,* and *WT1* displayed a significant level of overlapping in differentially methylated CpG sites (dmCpGs) [119]. Unsupervised hierarchical clustering allowed identifying six different epigenetic subclusters, associated with distinct mutations: Clusters 1 and 4 are mutated in *IDH1, IDH2, TET2*; cluster 2 in *DNMT3A*; cluster 3 in *DNMT3A-R882* and *WT1* [119]. Clusters 1, 3, and 5 displayed a significantly better survival than clusters 4 and 6 [119].

An extensive analysis of functional and topographic effects on DNA methylation in various *IDH1-IDH2* mutant cancers, including AML, showed that in addition to the previously described DNA hypermethylation phenotype, *IDH1-IDH2* mutant tumors display also a high level of DNA hypomethylation, which is particularly relevant at the level of gene promoters [120]. The CpG hypermethylator phenotype elicited by *IDH1-IDH2* mutations affects mainly non-promoter CpG islands with enhancer activity [120]. Interestingly, AML showed the most prominent hypermethylator phenotype and the highest level of tumor-specific hypermethylation [120].

## 10. IDH Mutations and AML Prognosis

Analyses of the prognostic impact of *IDH* mutations in AMLs emerged as a matter of great controversy, with contrasting evidence either supporting a positive, a negative, or a neutral impact on AML prognosis [121]. However, a recent study based on a large set of AML patients provided clear evidence that overall survival for *IDH*-WT AMLs and *IDH*-mutated AMLs as a whole is comparable [122].

In an attempt to better define the potential impact of *IDH* mutations on prognosis of various subtypes of AMLs, Xu et al. performed a large meta-analysis based on 33 published studies and reached the conclusion that: (i) *IDH* mutations seemed not to affect overall survival and event-free survival when considered as a single factor, but improved risk of relapse in patients with intermediate-risk karyotypes; (ii) *IDH1* mutation conferred worse overall survival and event-free survival, particularly in patients with normal cytogenetics; (iii) *IDH2* mutations confer a better prognosis in intermediate-risk AMLs, but not in patients with a normal karyotype [123].

There is growing evidence that the prognostic impact of *IDH* mutations in AML subtypes may be related to the co-mutational status. In this context, particularly interesting was the study of Amatangelo et al. [124]. These authors reported the co-mutation status in 125 *IDH2*-mutated AMLs undergoing treatment with the IDH2 inhibitor Enasidenib in the context of a phase I study. The co-mutation status clearly differed in *R140* and *R172*
*mIDH2* AMLs: *SFSR2* mutations were exclusively present in *mIDH2 R140* AMLs (45%), but absent in *mIDH2 R172* AMLs; *RUNX1* mutations were more frequent in *mIDH2 R140* than in *mIDH2 R172* AMLs (27% vs. 14%, respectively); *DNMT3A* mutations were more prevalent in *mIDH2 R172* than in *mIDH2 R140* AMLs (66% vs. 36%, respectively) [124].

The results of two clinical studies involving a total of 262 patients with *IDH1/IDH2*-mutated AMLs (101 *IDH1*^mut^, 115 *IDH2 R140Q^mut^*, and 46 *IDH2 R172^mut^*) treated with intensive chemotherapy were recently reported: *IDH1* mutations were significantly associated with *NPM1* and *DNMT3A* mutations, but mutually exclusive with *TET2* mutations; in these patients the association of *IDH1* mutations with *NPM1* mutations was linked to a better outcome, reinforced in the absence of *DNMT3A* mutations [125]. *IDH2 R140* mutations were significantly associated with *NPM1* mutations; the presence of concomitant *NPM1* mutations resulted in improved overall survival, reinforced by simultaneous absence of *DNMT3A* mutations, whereas concomitant *DNMT3A* mutations decreased overall survival time [125]. *IDH2 R172K* mutations were significantly associated with *DNMT3A* and *BCOR* mutations, as well as with +11 chromosomal abnormality, but negatively correlated with *NPM1* mutations; 78% of these patients achieved complete response, but not any specific genetic alteration was associated with the outcome [125].

In the revised European Leukemia Net classification of AMLs of patients aged <60 years were stratified among low-risk, intermediate-risk, and high-risk AMLs: In the low-risk AMLs, *IDH1* and *IDH2*-mutations are co-mutated with *NPM1* mutations: However, the presence of *IDH1* mutations was associated with a worse DFS and shorter OS; in the intermediate-risk AMLs, *IDH1* and IDH2 mutations are less frequently associated with NPM1 mutations and more frequently with *DNMT3A* and *FLT3-ITD* mutations; in the high-risk AMLs, *IDH1* and *IDH2* mutations are co-mutated with *DNMT3A* and *ASXL1*, and *IDH2* mutations only with *SRFS2* mutations: Patients harboring *IDH2* mutations had longer OS than patients with wild-type *IDH2* [126]. Thus, this revision proposes a reclassification of *IDH2*-mutant AMLs in the high-risk AMLs to the intermediate-risk AMLs because their outcome was similar to that of intermediate-risk AML patients [126].

*IDH* mutations promote the accumulation of the 2-hydroxyglutarate (2-HG) oncometabolite in the leukemic blasts and in serum/urine of *IDH* mutant AML patients. Thus, several studies have investigated the potential prognostic significance of 2-HG evaluation in AML patients. The 2-HG level evaluation in leukemic or serum/urine evaluation serves as a noninvasive biomarker of disease burden; serum 2-HG levels do not differ among *IDH1*- and *IDH2*-mutant AMLs [127]. In these patients, evaluation of serum 2-HG levels at remission had a prognostic value: Higher serum 2-HG levels in these patients predict an elevated risk of AML relapse [128]. This finding was confirmed by Janin et al., showing that the serum 2-HG level is a predictor of the presence of *IDH1/IDH2* mutations and outcome in these patients [129].

Wang et al. reported a screening of serum 2-HG levels in a large group of Chinese AML patients and showed that 17% of these patients displayed 2-HG levels above the cutoff value; 87% of AML cases with very high serum 2-HG levels displayed *IDH1* or *IDH2* mutations; 29% of patients with moderately increased serum 2-HG levels possessed IDH mutations, thus suggesting that events other than *IDH* mutations exist, causing an increased 2-HG production [130]. In AML patients with cytogenetically normal AMLs, high 2-HG was a negative prognostic factor in both overall and event-free survival [130].

Brunner et al. reported the study of 202 AML patients, whom 25% exhibited *IDH1* or *IDH2* mutations: These last patients displayed increased 2-HG levels in the serum, urine, or bone marrow [131]. A serum 2-HG level greater than 534 ng/mL was 99% specific for the presence of *IDH1-IDH2* mutations. The *IDH*-mutated AML patients showed an overall survival rate of 57% at two years following standard chemotherapy treatment; decreased serum 2-HG levels in these patients on day 14 of treatment were associated with improvements in overall survival and event-free survival [131].

The possible significance of pretreatment serum 2-HG levels in a group of 84 *IDH*-mutated AML patients [132]. The analysis of the entire IDH cohort of AML patients showed that serum 2-HG levels negatively impact event-free survival but had no effect on overall survival [132]. However, a subgroup analysis provided evidence that the negative effect of pretreatment serum 2-HG levels was restricted to AML patients bearing *IDH1* mutations [132].

## 11. Are IDH Mutations a Suitable Marker for Minimal Residual Disease?

The majority of patients with *de novo*-AMLs undergo morphological remission after standard chemotherapy treatment, but the rate of patients that relapse after this initial remission is high. The identification of a persistent small population of leukemic cells, called minimal residual disease (MRD) or measurable residual disease (MRD) is a key prognostic factor to evaluate the risk of relapse and represents a fundamental tool for evaluating post-remission therapy [133,134].

Klco et al. have performed a study on 71 AML patients treated with standard induction chemotherapy and showed that the detection of persistent leukemia-associated mutations in at least 5% of bone marrow cells in day 30 remission samples was associated with a marked risk of leukemia relapse and reduced overall survival [107]. Twelve of these AML patients displayed *IDH* mutations: 8/12 AML *IDH*-mutant AMLs at day 30 displayed a VAF < 2.5% and 4/12> 2.5%; all the *IDH*-mutant AML patients with a VAF > 2.5% pertain to the group of AML patients with a short event-free survival [107].

Jongen-Lavrencic et al. have carried out a fundamental study in 430 AML patients who achieved a complete response after standard chemotherapy [135]. Next generation sequencing studies showed that about 51% of these patients displayed persistent mutations at variable allele frequencies. Importantly, the detection of *DNMT3A, TET2,* and *ASXL1* mutations, which are present in individuals with ARCH, was not correlated with an increased relapse rate [136]. However, the persistence of mutations different from these three mutations during remission conferred significant prognostic value related to relapse [136]. Concerning *IDH1/IDH2* mutations, the frequency of *IDH1* and *IDH2*-mutant AMLs during remission markedly decreased; in cases with persistent *IDH1* or *IDH2* mutations the variant allelic frequency was low in about 2/3 cases and high (i.e., 20% or more) in about 1/3 of cases [136].

The level of mutation clearance at remission was highly variable for the various mutations: Mutations in *NPM1, CEBPA, FLT3,* and *NRAS* showed a high rate of mutations clearance, whereas *ASXL1, DNMT3A, TP53,* and *SRSF2* mutations displayed a low rate of mutation clearance; an intermediate condition was observed for the clearance of *IDH1* and *IDH2* mutations [109]. Rothenberg-Thurley et al. have analyzed the mutational profiling of 126 AML patients in pre-treatment and remission samples; 40% of these patients retained ≥1 mutation at remission with a VAF ≥2%; mutation persistence was most frequent in *DNMT3A* (65% of patients with mutations at diagnosis), *SRSF2* (64%), *TET2* (55%), *ASXL1* (46%), and *IDH1/IDH2* (30%) and was associated with reduced survival [112]. Patients without persistent mutations at diagnosis had an initial frequency of *IDH1/IDH2* mutations corresponding to about 16%; patients with persisting mutations at remission had an initial frequency of *IDH1/IDH2* mutations of about 34%; 53% of these *IDH* mutations persisted at remission and 47% disappeared [112].

Several studies have evaluated the utility of *IDH1* and *IDH2* mutations as suitable targets of MRD monitoring. Thus, Debarri et al. have reported the study of 31 AML patients with *NPM1* mutant AMLs harboring *NPM1* mutations in association with *IDH1/IDH2* or *DNMT3A* mutations: The monitoring of *IDH1/IDH2* mutations, but not *DNMT3A* mutations, allowed detecting the presence of leukemic disease and thus predicting relapse in the majority of patients [114]. Ferret et al. reported the study of 103 AML patients with *IDH* mutations enrolled on Acute Leukemia French Association (ALFA)-0701 and 0702 clinical trials. The mutant allele fraction (VAF) was 42.3% (range 8–49.9%) in bone marrow at diagnosis and below the detection limit of 0.2% (range <0.2-39.3%) in complete remission after induction therapy; in univariate analysis a *IDH1/IDH2* VAF < 0.2% in bone marrow after induction therapy was a predictor of longer disease-free survival [137]. In 7% of patients, *IDH1/IDH2* mutations persisted at high levels in complete remission, suggesting the presence in these patients of *IDH* mutations at the level of the preleukemic stem cell pool; five out of these seven patients relapsed or progressed to MDS [137].

Ok et al. reported the study of 80 AML patients with *IDH* mutations, associated with *IDH* mutations at the time of the remission after induction therapy: About 40% of these patients had persistent *IDH* mutations, associated with an increased risk of relapse after one year of follow-up compared to patients without a detectable *IDH1/IDH2* mutation (59% vs. 24%, respectively) [138]. However, in spite this finding, a high *IDH1/IDH2* burden did not correlate with relapse rate [138].

The persistence of *IDH1/IDH2* mutations is a predictor of relapse also in AML patients undergoing allogeneic stem cell transplantation [139]. Gotta et al. have investigated the clearance of leukemia mutated alleles between diagnosis and before hematopoietic cell transplantation and observed that mutations in *DNMT3A, TET2,* and *JAK2* were less likely to be cleared than *NPM1, IDH1/IDH2,* and *FLT3-ITD* [140]. Particularly, *IDH1* and *IDH2* mutations were cleared in nine of 11 cases and VAF decreased from 25% to 3% [140]. The presence of flow cytometry minimal residual disease and persistent leukemic mutations before hematopoietic stem cell transplantation was associated with relapse risk and reduced survival [140]. Duncavage et al. have explored the mutation profile of 90 adult patients with MDS who underwent allogeneic hematopoietic stem cell transplantation after a myeloablative or a reduced intensity conditioning regimen: The frequency of *IDH2*-mutant AMLs was higher among patients with progression post-transplantation (14%) compared to that observed in patients without progression (5.8%) [141].

Interestingly, Ediriwickrema et al. in a very recent study have investigated MRD at single-cell level [142]. Thus, single-cell sequencing was used to evaluate the clonal dynamics of AML from diagnosis to remission and to relapse. In this study, MRD was considered as the expansion of clones observed at remission that enlarged into the dominant clone at relapse [142]. Particularly, single-cell sequencing detected and quantified both pre-leukemic clonal hematopoiesis clones and leukemic clones that may become dominant at relapse [142]. According to this study, the mutations were classified as early or late: *DNMT3A* and *IDH2* were classified as early mutations; however, in some patients, *IDH2* mutations seem to be acquired at later stages [142].

The definition of a reliable biomarker of molecular MRD status is of fundamental importance to predict the response of AML patients to HSC transplantation. *NPM1* mutations represent the most suitable molecular biomarker for *NPM1*-mutant AMLs. The long-term results of the National Cancer Research Institute AML17 study have been recently published: After a median follow-up of about five years, patients with negative, low, and high levels of MRD had a two-year survival of 83%, 63%, and 13%, respectively [143]. The analysis of patients with low-level MRD, showed that patients with *FLT3-ITD* mutations had poorer outcome [143].

In conclusion, these studies indicate that persistent mutations after therapy in remission samples are associated with an increased risk of relapse; however, additional studies based on large cohorts of patients are required to demonstrate that the residual level of *IDH1/IDH2* mutations at remission may support the assignment of patients into distinctly risk subgroups and guide the choice of optimal therapy (i.e., allogeneic stem cell transplantation or targeted therapies).

## 12. IDH Inhibitors

As above discussed, several observations strongly supported a role for mutant *IDH* as a valuable therapeutic target. In fact, *IDH1/IDH2* mutations are recurrent in AMLs and are responsible for the development of a leukemic phenotype, characterized by the production of an oncogenic metabolite, responsible for a block of cell differentiation and changes in gene expression. Preclinical studies have shown that all these changes can be reversed by inhibition of expression/activity of mutant enzymes. These observations have strongly supported the development of specific IDH inhibitors suitable for clinical studies. Thus, the mutant *IDH2* inhibitor AG-221 (Enasidenib) and mutant *IDH1* inhibitor AG-120 (Ivosidenib) have been extensively investigated for the treatment of patients with AML or MDS with a susceptible *IDH* mutation.

### 12.1. IDH2 Inhibitors

AG-221 (enasidenib) is an orally available, potent, and specific triazine inhibitor of the mutant *IDH2* [144]. A high-resolution X-ray crystal structure of enasidenib in complex with IDH2^R140Q^ showed that this drug binds to the allosteric site enclosed within the homodimer interface and thus induces an open, inactive conformation of the mutant enzyme [143]. Preclinical studies have shown that this inhibitor markedly reduced 2-HG levels in multiple leukemia models, including tumor xenograft models [143]. Enasidenib induced differentiation of various types of leukemic cells expressing mutant *IDH2*, such as mutant *IDH2* TF-1 erythroleukemic cells, primary human AML cells ex vivo, four *IDH2-R140Q*-mutant human AML xenografts mouse models in vivo [143]. Furthermore, enasidenib improved the survival in vivo in an aggressive human AML xenograft model [143]. These observations strongly supported the clinical use of enasidenib.

The availability of a potent IDH2 inhibitor was of fundamental importance for the development of preclinical models attempting the combined inhibition of IDH2, together with other altered genes or pathways. Thus, Shih et al. have reported the development of a combination therapy based on the study of a mouse model generated using combined transformation with mutant *TET2, FLT3-ITD,* and *IDH2-R140Q*; thus, triple-transformed leukemia resulted to be sensitive to 5-azacytidine or to the IDH2 inhibitor enasidenib [145]. The combined treatment with these two drugs resulted in a marked potentiation of the antileukemic effect, with a pronounced decrease of leukemic blasts and with their differentiation and, particularly, with a decrease of mutant allele burden and progressive recovery of normal hematopoiesis from non-mutant stem-progenitor cells [145].

A fundamental phase I/II clinical trial in 2017 generated the first clinical data on the safety profile and therapeutic efficacy of enasidenib in the treatment of *IDH2* mutant AML with relapsing or refractory disease [146] (Table 2). The dose-escalation phase of the study showed that enasidenib was tolerated up to the maximum dose of 650 mg daily; 10 mg was the dose selected for the expansion phase of the study, involving the enrollment of 101 AML patients [146]. Seventy six percent of these patients displayed *IDH2-R140* mutations and 24% *IDH2-R172* mutations [146]. The overall response rate of these patients was 40.3%, with about 20% of patients achieving a complete response [146]. The objective response rate was higher among patients with *IDH2-R172* mutations than in those with *IDH2-R140* mutations (24% vs. 17.7% of complete responses, respectively). The mean duration of response was 5.8 months and median overall survival was 9.3 months, with an estimated one-year survival of 39%; in patients achieving a complete response, the median overall survival was 19.7 months; finally, 10% of these patients were bridged to stem cell transplantation [146]. The most common adverse events were hyperbilirubinemia, thrombocytopenia, anemia, and IDH differentiation syndrome [146]. The results of this trial have led to the FDA accelerated approval for the use of enasidenib for the treatment of *IDH2*-mutated relapsed/refractory AMLs [146].

Stein et al. have reported an update of their initial study, showing the rate and the pattern of molecular remission in 214 patients with relapsed or refractory *IDH2*-mutant AMLs: The overall response rate was 38.8%, with 19.6% of these patients attaining complete response and 10.3% proceeding to allogeneic bone marrow transplantation [147]. Response rates were similar for patients in relapse or with refractory disease and for patients with either *IDH2-R140* or *IDH2-R172* mutations [147]. The extent of 2-H-HG levels following the treatment with enasidenib was associated with complete remission in *IDH2-R172* patients [147]. Forty three percent of RBC transfusion-dependent and 40% of platelet transfusion-dependent patients become transfusion-independent [147]. Variant allele frequency (VAF) was analyzed in these patients before and after treatment with enasidenib, showing that 12% of patients, all with *IDH2-R140* mutation, achieved molecular remission during enasidenib treatment: Of these patients, the large majority (>80%) achieved complete remission [147]. Importantly, among all patients who achieved a complete response, survival was similar for both those who attained molecular remission and those who did not [147]. De Botton et al. very recently provided a new evaluation of the data on the effect of enasidenib on overall survival of refractory/relapsing AML patients, providing support to the view that this IDH2 inhibitor may prolong the survival of patients with refractory/relapsing mutant *IDH2* AML, compared to the standard of care [148].

Correlative studies have been carried out on samples derived from the patients enrolled in this large clinical trial on mutant *IDH2* AML patients and several observations have been made helping to better understand the mechanisms through which enasidenib exerts its antileukemic effects [123,147]. Thus, a fundamental observation was made that the mutant allelic burden was found to be identical in mature neutrophils as well as in immature leukemic stem/progenitor cells, thus suggesting that the main effects of enasidenib consist of inducing leukemic cell differentiation and not leukemic cell apoptosis; enasidenib induces a marked reduction of R-2-HG levels; co-occurrence of mutations at the level of MAPK and RAS pathways was associated with reduced clinical response to enasidenib [123,147]. Stein et al. provided a detailed analysis of *IDH2*-mutant relapsed/refractory AML patients treated with enasidenib. The analysis of mutant-*IDH2* VAF showed for patients with *IDH2-R140* mutations a marked decline, reaching 90% reduction by cycle 9 of treatment and being more pronounced in patients achieving complete remission, compared to nonresponders; the VAF decrease in patients with *IDH2-R172* mutations was less pronounced [149]. In spite of these differences in the decline of mutant VAF among *IDH2-R140* and *IDH2-R172* mutants, their clinical response to the enasidenib treatment was comparable [148]. About 12% of *IDH2-140* patients treated at enasidenib dosage of 100 mg achieved mutant-*IDH2* molecular remission, defined as the IDH2 mutant VAF below the detection limit; molecular remission was associated with achievement of hematological response and with improved overall survival (22.9 vs. 8.8 months) [148]. The analysis of the co-mutation profiles of the various *IDH2*-mutant patients provided evidence that responding patients had significantly less baseline mutations than nonresponding patients; *NRAS* and *FLT3* co-mutations were associated with a significantly reduced rate of response to the enasidenib treatment [148]. Among patients with *IDH2-R140* mutations, complete responders displayed a frequency of *NRAS, FLT3,* and *NPM1* mutations of 5%, 5%, and 26%, respectively and in non-responders of 15%, 18%, and 12%, respectively [150]. Importantly, response rates were similar among patients who were in relapse or were refractory to standard therapies [150].

The differentiation syndrome is one of the adverse events most frequently associated with the enasidenib treatment. Fathi et al. have explored the occurrence of differentiation syndrome in a cohort of 291 relapsed/refractory *IDH2*-mutant AML patients treated with enasidenib and observed a frequency of 11.7% [151]. The most common signs of differentiation syndrome were dyspnea, unexplained fever, pulmonary infiltrates, and hypoxia [151]. About 30% of patients with differentiation syndrome required intensive care unit admission [151].

Pollyea et al. reported the results of a phase I/II clinical study involving the treatment of 39 older *IDH2*-mutant AML patients, who were not candidates for cytotoxic regimens [152]. About 30.8% of patients displayed a response, of whom 18% displayed a complete response [153] (Table 2). The median overall survival for all patients was 11.3 months [152]. Response rates were moderately different in relation to the co-mutational burden at study entry, with an overall responding rate of 47% in patients with ≤ 3 co-occurring mutations at baseline, compared to 27% among patients with ≥ four baseline co-mutations [152]. At the level of individual co-mutations, the presence of a *DNMT3A* mutation was significantly associated with complete clinical responses; in line with this finding, patients with co-mutations at the level of genes involved in DNA methylation pathway showed a trend toward achievement of a complete response [152].

As above discussed, preclinical studies have supported the rationale of the association of azacitidine with enasidenib. Di Nardo et al. have reported the preliminary results of a clinical phase II study (NCT 02677922) involving the treatment of 101 adult *IDH2*-mutant AML patients ineligible to receive intensive chemotherapy, who have been randomized in a 2:1 ratio to receive enasidenib + azacitidine or azacytidine alone [154] (Table 2). The mean age of treated patients was comprised between 74 and 75 years. Response rates were significantly higher among patients treated with the combination treatment, compared to those observed with azacytidine alone: Overall response rates were 68% vs. 42%, respectively and complete response rates were 50% vs. 12% [154]. Importantly, the duration of response was not reached in the combination arm, whereas it was of 10.2 months in the azacytidine-only arm; furthermore, the suppression of the mutant *IDH2* allele was much greater with combination treatment (-69.3%) than with azacytidine alone (−14.1%) [154]. In spite of these remarkable differences in the complete remission rates, the overall survival was not significantly improved with the combination therapy compared to single-agent azacytidine alone; however, 21% of patients received subsequent enasidenib following treatment discontinuation and this may account for the lack of difference in overall survival data [154]. These results strongly support the rationale of using a combination treatment based on a hypomethylating agent and an IDH2 inhibitor for the treatment of *IDH2*-mutant AML patients.

Despite an initial response, some AML patients relapse and develop resistance to an additional treatment with enasidenib. Recent studies have explored at molecular level the possible mechanisms of this acquired drug resistance. Intlekofer et al. have explored two patients with *IDH2*-mutant AML who had an initial clinical response to enasidenib and then developed disease recurrence, associated with resistance to additional treatment with enasidenib [155]. During the initial response to treatment, enasidenib promoted differentiation of *IDH2*-mutant blasts, in association with an unmodified variant allele frequency for the *IDH2-R140Q* mutation [155]. At the time of recurrence, in both patients, new mutations in the *IDH2* gene were identified, corresponding to a Q316E mutation in the first patient and I1319M mutation in the second patient [155]. These mutations were undetectable in the initial leukemic samples [155]. Interestingly, these second mutations occurred at the level of the *IDH2* allele initially not affected by the mutation [155]. When tested in recipient cells, these second mutations alone did not affect the IDH2 function, but when co-expressed with *IDH2-R140Q* induced resistance to the enosidenib treatment. Molecular modeling indicated that these second mutations did not affect the catalytic site and seemingly determined a perturbation of the dimer interface at the level of the site of interaction between enasidenib and its target [156]. The second-site mutations were detected in two of nine cases of acquired resistance and were not observed in 14 cases displaying primary resistance [155].

A second study identified a different mechanism underlying the development of acquired resistance to the enasidenib treatment. Thus, Quek et al. have analyzed the clonal composition of *IDH2* mutant AMLs and showed that in pre-treatment samples the capacity of enasidenib to induce leukemic cell differentiation is mainly related to the type of co-associated mutations present in each single clone. The various clones are very heterogeneous in their response to enasidenib: In the majority of cases, the differentiated clone was either the ancestral or the terminal clone [157]. In the 16 patients investigated in this study, 2-HG levels remained markedly decreased at relapse in 14 cases: In these 14 cases, relapse was originated by clonal evolution with acquisition of new mutations (such as *RUNX1* and *FLT3*) at the level of new driver genes or by selection of pre-existing clones; these developing clones conferring resistance were represented by ancestral clones, while in other instances by presenting clones [157]. Very interestingly, in two patients rising 2-HG levels and BM blasts were observed in spite of enasidenib administration: In these patients, a different mechanism of resistance was observed and represented by the acquisition of IDH1 mutations (*IDH1-R132C/H*), not detectable by pre-enasidenib therapy. Finally, in this study, no second-site *IDH2* mutations were observed in patients acquiring resistance.

### 12.2. IDH1 Inhibitors

Ivosidenib, initially known as AG-120, is a small-molecule, specific inhibitor of mutant *IDH1*, displaying inhibitory activity of a variety of *IDH1-R132* mutants at concentrations much lower than those required for inhibition of normal *IDH1* [158]. Preclinical studies have supported the capacity of ivosidenib to exert an inhibitory activity on mutant IDH1-AML cells, as shown by: (i) Low concentrations of ivosidenib are sufficient to inhibit 2-HG levels in mutant *IDH1* leukemic cell lines; (ii) treatment of primary mutant *IDH1* leukemic blasts with ivosidenib reduces 2-HG production, induces cell differentiation, and reduces cell viability; (iii) in a xenograft model generated with primary human AML cells bearing an *IDH1-R132H* mutation, ivosidenib markedly decreased 2-HG levels in vivo and induced leukemic cell differentiation [158].

These preclinical studies have strongly supported clinical trials aiming to evaluate the safety profile and the antileukemic activity of ivosidenib in mutant *IDH1* AMLs. In this context, fundamental was the clinical study AG120-C-001, a multicenter, open-label, single-arm, dose-escalation trial based on the administration of ivosidenib to patients with advanced hematologic malignancies harboring an IDH1 mutation [159] (Table 3). In a population of 179 refractory/relapsing AML patients the rate of complete remissions + complete remissions with partial hematological recovery was 33%; a trend toward lower response rates was observed among patients with poor risk cytogenetics, prior to hematopoietic stem cell transplantation and two or more prior to therapies and the *R132H* mutation [159]. Among patients who achieved complete remission the median duration of response was 10.3 months [159]. The treatment with ivosidenib was well tolerated in these patients, with an incidence of differentiation syndrome of 19% (13% of grade 3 or higher) [159].

Roboz et al. reported the results relative to 34 newly diagnosed older adult AML patients from a phase I study enrolling patients with *IDH1*-mutant hematologic malignancies at advanced stage (Table 3). These patients had a median age of 76 years, were ineligible for standard therapies and received 500 mg of ivosidenib once daily; furthermore, 76% of these patients had secondary AML and 47% were previously treated with at least one hypomethylating agent [160].

The complete remission rate, including also complete remission with partial hematologic recovery was 42%; 78% of patients with complete remission remained in remission at 1 year; with a median follow-up of 23.5 months, median overall survival was 12.6 months [160]. The most common adverse events were diarrhea, fatigue, nausea, and decreased appetite; differentiation syndrome was observed in 18% of cases and in 50% of these cases did not require treatment discontinuation [160]. Of 63% transfusion-dependent patients at baseline, 43% became transfusion independent. Importantly, *IDH1* mutation clearance was observed in about 65% of patients achieving a complete response and in none of the patients who did not achieve complete remission [160]. No single gene mutation was significantly associated with complete responses; however, receptor tyrosine kinase pathway mutations were not observed in any of the patients achieving complete response to the treatment with ivosidenib, whereas were detectable in 37% of patients who did not achieve complete remission [160].

Recent studies have explored the mechanisms of resistance to ivosidenib. There are two types of resistance: A primary resistance, corresponding to those patients not responding to the initial treatment with ivosidenib; a secondary resistance, corresponding to those patients initially responding to treatment, but after a variable time relapsing with a resistant disease. A first study provided evidence about a peculiar phenomenon of mutant IDH isoform switching as a key mechanism of development of resistance to IDH inhibitors. Particularly, it was identified as mutant IDH isoform switching, either from cytoplasmic mutant *IDH1* to mitochondrial mutant *IDH2* or vice versa, as a mechanism of acquired clinical resistance to IDH inhibition [135]. Four cases were reported to support this conclusion: (i) Two patients with relapsed/refractory *IDH1-R132C*-mutant AML who achieved remission in response to ivosidenib, followed by leukemic progression on therapy, rise of 2-HG levels, and emergence of *IDH2-R140Q* mutations; (ii) the third case concerns a patient with treatment-refractory *IDH1-R132*-AML who attained a partial remission following ivosidenib treatment, followed by disease progression with acquisition of the new *IDH2-R172V* mutation; (iii) finally, the fourth patient had a relapsed/refractory *IDH2-R140Q*-mutant AML who initially achieved a durable remission following treatment with enasidenib, with emergence of a new *IDH1-R132C* mutation [135]. These observations suggested the existence of a selective pressure to maintain 2-HG production in *IDH*-mutant AMLs and suggest also strategies to bypass these drug resistances using sequential administration of IDH1 > IDH2 or IDH2 > IDH1 inhibitors [135]. A second study by Wang et al. explored 174 refractory/relapsed AML patients with *IDH1*-mutant AML treated with ivosidenib: 129 patients responded to the treatment, whereas 45 were resistant; mutant *IDH2* was detected in 15 patients during treatment [161]. Single-cell mutational profiling showed multiple evolutionary mechanisms by which mutant *IDH2* contributes to relapse: In the majority of patients, resistance evolves through acquisition of mutant *IDH2* within mutant *IDH1* clones; in a minority of cases, in patients with the mutant clone being cleared with ivosidenib treatment, two possible evolutionary mechanisms have been identified: (i) Evolution of *IDH*-wild type clone; (ii) expansion or evolution of multiple mutant *IDH2* clones [161]. In all these patients, the acquisition of mutant *IDH2* as a secondary event is associated with restoration of 2-HG production [161]. In a third study, Choe et al. have explored the molecular mechanisms responsible for primary and secondary resistance to ivosidenib occurring in refractory/relapsed mutant *IDH1* AML patients [162]. In this study, the role of both pre-therapy (pre-therapy genetic profile predicts response) and post-therapy (the acquisition of new mutations through an IDH-dependent and IDH-independent mechanisms) was explored [162]. Thus, a longitudinal next generation sequencing analysis was performed on 105 *IDH1*-mutant AML patients treated with ivosidenib. In line with previous studies, the most frequent co-occurring mutations at baseline are represented by *DNMT3A* (about 35%), *NPM1* (about 26%), *SRSF2* (about 24%), *RUNX1* and *ASXL1* (about 18%), *NRAS* (about 14%), TP53 (about 13%) and other less frequent mutations; *IDH2* mutations were observed in 2% of these patients [162]. A significant association between RTK pathway (*FLT3, KIT*) mutations, as well as *NRAS* and *PTPN11* mutations, and lack of complete responses was observed; patients with *JAK2* mutations achieved a higher rate of complete remissions compared to *JAK2-WT* patients (64% vs. 32%, respectively) [162]. Interestingly, the variation of allelic frequency of mutant *IDH1* mutations is frequently high when associated with RTK pathway mutations, while the contrary when *IDH1* mutations are associated with co-mutations typically related to clonal hematopoiesis and or myelodysplasia [162]. Patients with a clonal or subclonal pattern of mutant *IDH1* exhibit the same frequency of complete remissions to ivosidenib therapy [162]. At relapse, IDH-related mutational events are frequently observed at the level of the IDH pathway, with 15% of *IDH2-R140Q* mutations and 15% of *IDH1*-second site mutations; *IDH1* second-site mutations were detected at relapse or disease progression, with concurrent 2-HG increases; functional studies showed that *IDH1* second-site mutations exhibited a decreased sensitivity to ivosidenib [162].

A very recent study evaluated the therapeutic impact of ivosidenib with azacitine. Thus, Montesinos Fernandez et al. recently reported the preliminary results of an ongoing phase I study (NCT 02677922) on 23 older (median age 76 years) newly diagnosed AML patients treated with ivosidenib (500 mg once daily) in combination with subcutaneous azacitidine [163] (Table 3). The overall response rate was 78%, including about 61% of complete responses; median response duration has not been reached [163]. Clearance of mutant *IDH1* allele in bone marrow mononuclear cells was observed in 69% of patients with complete response [163]. The analysis of 23 patients enrolled in this study showed a co-mutation pattern involving *RUNX1* (35%), *SRSF2* (35%), and *DNMT3A* (20%) as most frequently co-mutated genes [164]. Longitudinal analysis of mutation clearance in these patients using a sensitive technique, such as digital PCR assay, showed that there is a good concordance between *IDH1* mutation clearance and clinical response, evaluated as complete response rates [164]. Interestingly, the mutational clearance rates observed in these patients were higher than those previously reported in patients treated with ivosidenib alone [164]. In patients achieving complete remission, all mutations were cleared in 79% of the patients, apart from mutations in the genes associated with clonal hematopoiesis [164]. On the basis of these findings, a phase III double-blind placebo-controlled study of ivosidenib plus azacytidine (AGILE, NCT 03173248) is actually enrolling patients (Table 3).

## 13. Bcl-2 Targeting

Recent studies have supported potential clinical benefit deriving from the use of the BCL-2 (B-cell lymphoma-2) inhibitor venetoclax in combination with the hypomethylating agents or with low-dose cytarabine (LDAC) for the treatment of older AML patients, ineligible for intensive chemotherapy.

The most responsive patients to the treatment with venetoclax + LDAC are those with *IDH1/IDH2* (with a median overall survival of 19.4 months) and *NPM1* mutations, and for the treatment with venetoclax + hypomethylating agents are also those with *IDH1/IDH2* (with a median overall survival of 24.4 months) and *NPM1* mutations [165]. The characterization of 81 older patients undergoing treatment with venetoclax + LDAC or azacytidine and showed that high-response rates are associated with *NPM1* or *IDH2* mutation, while primary and adaptive resistance is related to *FLT3* or *RAS* mutations or biallelic alteration of *TP53* [166]. Particularly, in this study the mutational profile was explored in three subgroups of patients subdivided according to the response to the venetoclax treatment: Durable remission, remission, then relapse and primary refractory [166]. Two gene mutations, *NPM1* and *IDH2* mutations, were considerably enriched among patients achieving a durable remission; these two gene mutations were much less frequent in patients who achieved only a transient remission (12% for *NPM1* and 4% for *IDH2*) and in those who were refractory to treatment (5% for *NPM1* and 0% for *IDH2*) [166]. Two *IDH2*-mutant AMLs achieving durable remissions display IDH2 mutations either in association with NPM1 mutations or with *RUNX1* mutations [166]. In contrast, the frequency of *IDH1* mutations was not associated with the response to venetoclax: The frequency of *IDH1*-mutant AMLs was 12% in patients achieving durable response, 20% in patients with transient relapse, and 10% among refractory patients [166].

The CAVEAT trial, involving the initial treatment with seven-day administration of venetoclax, followed by venetoclax in combination with chemotherapy provided preliminary evidence that *IDH1/IDH2* mutant AMLs are sensitive to this treatment. Thus, *NPM1*-mutant and *IDH1/IDH2*-mutant AMLs are the AMLs achieving greatest bone marrow blast reductions after seven days of venetoclax monotherapy; clinically complete remission and complete remission with incomplete count recovery were observed in 100% *IDH2*-mutant AMLs and in 62% of *IDH1*-mutant AMLs [167]. Median survival for *IDH2*-mutant AMLs was not reached [166]. Variation allele frequency decreased in 55% of *IDH1/IDH2*-mutant AMLs [167]. These observations suggest that the treatment naïve *IDH2*-mutant AML blasts are highly sensitive to venetoclax in combination with cytarabine and anthracycline chemotherapy, resulting in a high clinical response rate; in contrast, *IDH1*-mutant AMLs seem to be less sensitive to this treatment [167].

Chyla et al. have reported the results on the clinical outcomes of a group of older AML patients (median age 74 years) undergoing treatment with venetoclax + hypomethylating agents or low-intensity chemotherapy, correlating clinical responses with molecular markers and with the levels of BCL-2 expression [168]. The percentages of patients who exhibit a complete response or a complete response with incomplete hematological recovery was high among *IDH1/IDH2*-mutant (83.7%) and *NPM1*-mutant (84.6%) AMLs [168]. The median overall survival was not reached for *IDH1, IDH2,* or *NPM1* mutant AMLs, with a median time in study of 11.6 months (range from 0.3 to 44 months) [168]. Finally, patients with *IDH1/IDH2*-mutant AMLs show a tendency to have higher BCL-2 mRNA levels than other AML subsets [168].

The response to combination therapy with hypomethylating agents and venetoclax is limited in patients with relapsed/refractory disease. Ashgari et al. have investigated the outcomes in a group of 72 patients with relapsed/refractory AMLs [169]. The results of this study confirmed initial observations, showing that the hypomethylating agent and venetoclax combination as salvage setting is of limited efficacy, with the possible exception of AML patients with *IDH1/IDH2* mutations [169].

Given the efficiency of IDH inhibitors and venetoclax in *IDH1/IDH2*-mutated AMLs it is expected that combinations of these drugs will act synergistically in patients harboring these mutations. This view is currently under evaluation in a clinical trial based on the administration of venetoclax and ivosidenib, with or without azacytidine in refractory/relapsed AML patients with *IDH1*-mutant AMLs (NCT 03471260) (Table 2). Preliminary results based on 12 patients treated with ivosidenib and venetoclax have shown a rate of complete responses of 75% which compares favorably with response rates of about 40% observed with ivosidenibin refractory/relapsed *IDH1*-mutant AMLs [170]. Importantly, in these 12 patients no signals of significant added toxicity have been observed with the combination of these two drugs, thus suggesting triplet-drug combinations including azacitidine could have a favorable risk-benefit profile.

The exploration of the mechanisms of venetoclax resistance occurring in some AMLs has led to clarify the mechanisms underlying the peculiar sensitivity of IDH-mutant AMLs to venetoclax [144]. Repression of *TP73* in *IDH1/IDH2*-mutant AML and downregulation of *TP73* by the oncometabolite 2-HG were associated with enhanced sensitivity to venetoclax, thus supporting the view that TP73 determines AML susceptibility to BCL-2 inhibition; in contrast, venetoclax resistant AML cells overexpress *TP73* and *TP73* knockdown in these cells restores venetoclax resistance [144]. Azacitidine determines a decrease of TP73 levels and improves the anti-leukemic effect of venetoclax [144].

## 14. IDH Inhibitors in Myelodysplastic Syndromes

Stein et al. have recently reported initial observations on 17 patients with relapsed/refractory mutant *IDH2* MDS, enrolled in the AG 221-C001 trial (Table 1). At the entry, 18% of patients had relapsed after allogeneic stem cell transplantation, 76% had previously received therapy with hypomethylating agents, and 59% had previously received at least two therapies; all these patients have been treated with enasidenib [171]. An overall response was observed in 53% of patients with a median duration of response of 9.2 months; 46% of 13 patients previously treated with hypomethylating agents responded to the enasidenib treatment; median overall survival was 16.9 months and median event-free survival was 11 months [171].

Richard-Carpentier et al. reported the preliminary results of a phase II study designed to evaluate the efficacy and tolerability of enasidenib alone and in combination with azacytidine in patients with high-risk *IDH2*-mutated MDS [172]. The study included two cohorts of patients: Hypomethylating agent-naïve patients with high-risk MDS treated with enasidenib plus azacytidine; relapsed/refractory high-risk MDS patients previously treated with hypomethylating agents received enasidenib alone [172]. Among the 18 patients available, the overall response rate was 67%: In hypomethylating agent-naïve patients, 100% responded to the treatment; in the other group of patients, 50% responded [172]. Three patients who achieved complete response also had clearance of the *IDH2* mutation [172].

Another recent study reported the results observed in a small cohort (12 patients) of *IDH1*-mutant MDS patients with relapsed/refractory disease treated with 500 mg ivosidenib once daily. The treatment was well tolerated and five of twelve patients achieved complete response and nine of twelve patients were transfusion independent during a part of the treatment [173]. Furthermore, mutation clearance was observed in one of the five patients achieving a complete response [173].

## 15. New IDH Inhibitors under Evaluation

Recently, the identification of a new potent, mutant-selective IDH1 inhibitor FT-2102 (Olutasidenib) was reported: This inhibitor is highly active in mutant IDH1 xenotransplantation mouse models, orally bioavailable, with excellent pharmacokinetic properties, and is a promising candidate for treatment of hematologic, solid, and brain tumors [149].

Watts et al. recently reported the results of a phase I clinical study based on the administration of olutasidenib alone or in combination with azacitidine in *IDH1*-mutant AML patients with relapsed/refractory disease or treatment-naïve not eligible for standard therapy. Clinical responses were observed in 39% of patients treated with monotherapy with olatusidenib (with 15% of complete responses) and in 54% of patients treated with the combination regimen (with 23% of complete responses); about 40% of the treated patients became transfusion-independent; for refractory/relapsed AML patients the median overall survival was 8.8 months for monotherapy and 12.1 months for combination treatment; for treatment-naïve AML patients the median overall survival for single-treatment was 8.7 months and not reached for the combination treatment; *IDH* mutant clearance or significant reduction was observed in 40% of patients achieving an objective response [174].

Cortes et al. reported the preliminary results of a phase I/II clinical trial involving the treatment of 20 *IDH1*-mutant MDS patients with olutasidenib alone or in combination with azacitidine; clinical responses were observed in 33% of patients treated with monotherapy and in 73% of patients treated with combination therapy [153].

A recent study reported the characterization of HMS-101 a new IDH1 inhibitor with the peculiar property to interact with the active site of *IDH1*-mutant in close proximity to the regulatory segment of the enzyme [175]. The inhibitor exerted a potent anti-leukemic effect on mutant *IDH1* leukemia models; interestingly, leukemic cells treated with this inhibitor showed a marked upregulation of the transcription factors CEBPA and PU.1 and a decrease of cyclin A2 [175].

## 16. PARP Inhibitors Are Effective in IDH1/IDH2 Mutant AML and MDS

2-HG accumulation occurring in *IDH*-mutant leukemic cells inhibits the function of histone demethylases, such as KDM4A and KDM4B, whose activity is essential for homologous recombination (HR) DNA repair pathway and, therefore, for the repair of DNA double strand breaks (DSBs) [176]. Thus, Sulkowski et al. showed that *IDH1/IDH2* mutations induce a HR defect that renders cancer cells exquisitely sensitive to poly (adenosine 5′-diphosphate-ribose) polymerase (PARP) inhibitors [176]. This “BRCAness” phenotype is dependent on the mutant *IDH* in tumor cells since it is completely reversed by inhibition of mutant *IDH* allele; furthermore, 2-HG completely recapitulates the effects induced by mutant *IDH* on PARP sensitivity [176]. These findings have provided the rationale for the development of a possible new therapeutic strategy based on the targeting of 2-HG-dependent HR deficiency using PARP inhibitors [176].

In *IDH* mutant tumors PARP inhibitors induce synthetic lethality by repressing the repair of DNA single strand-breaks, which are converted into DSBs [156]. Molenaar et al. have explored the levels of DNA damage and sensitivity to PARP inhibitors and DNA damage-inducing chemotherapy in mutant *IDH1*, mutant *IDH2,* and IDH-WT AML cells [156]. The results of this study showed that primary *IDH1/IDH2*-mutant AML cells have reduced DNA damage responses and reduced expression of ATM [156]. As a consequence of these defects, *IDH*-mutant AML cells are sensitive to PARP inhibitors as monotherapy, but particularly when combined with a DNA-damaging agent, such as daunorubicin; in contrast, concomitant administration of IDH1/IDH2 inhibitors during cytotoxic therapy decreases the efficacy of PARP inhibitors or of daunorubicin [156].

A recent study explored in more detail the spectrum of sensitivity of IDH-mutant AMLs to PARP inhibitors. Thus, the study of two syngeneic mouse models of MDS and AML based on co-mutation of *IDH2/SRSF2* or *IDH2/FLT3* further supported the sensitivity of *IDH2*-mutant AML cells to PARP inhibitors [177]. Importantly, leukemic cells bearing *IDH2*-mutations resistant to IDH2 inhibitors are sensitive to the PARP inhibitor olaparib [177]. Olaparib pretreatment of IDH-mutant MDS or AML cells displayed a marked reduction of their engraftment capacity, thus suggesting an inhibitory effect of PARP inhibitors on leukemic-initiating cells [177]. These observations support the conclusion that PARP inhibitors are effective in vivo against *IDH2*-mutant MDS and AML and are able to overcome targeted IDH inhibitor resistance [177].

These observations strongly supported the development of clinical trials aiming to evaluate the potential clinical benefit deriving from the treatment of IDH1/IDH2 mutant AML and MDS with clinically approved PARP inhibitors. In this context, the PRIME trial (NCI10264) was recently proposed, as a proof of concept, biomarker-driven, phase II clinical trial to evaluate the overall response of IDH1/IDH2-mutant refractory/relapsed AML and MDS to PARP inhibitor monotherapy with olaparib [178].

It is of interest to note that various leukemia-driven oncogenes, such as *IDH1/IDH2, TET2, PML-RARA, TCF3-HLF,* and *RUNX1-RUNXT1*, or treatment with targeted agents directed against aberrant kinases, such as FLT3 and JAK1/2 inhibitors, have been linked to reduced DNA repair activity, a condition that renders leukemic blasts sensitive to PARP inhibitors [179].

## 17. Conclusions

AML is a heterogeneous disease, characterized by a broad spectrum of molecular alterations; some of these alterations are key driver events of the leukemic process and influence clinical outcomes. Mutations in epigenetic modifiers, including *IDH, DNMT3A, TET2, ASXL1,* and *EZH2* frequently occur in patients with AML. The successful development of effective targeted therapies for *IDH1*- and *IDH2*-mutant AMLs has led to the regulatory approval of IDH1 and IDH2 inhibitors, improving response rates and outcomes for patients whose leukemia harbors these mutations. Thus, in 2018 the FDA approved ivosidenib and enasidenib for the treatment of patients with relapsed or refractory *IDH1*-mutant and *IDH2*-mutant AMLs, respectively. Subsequently, in May 2019 single-agent ivosidenib was approved also for the treatment of older (≥75 years of age) AML patients or for patients unfit for intensive chemotherapy. Unfortunately, not all patients with IDH-mutated AMLs respond to IDH inhibitors and a significant proportion of responding patients develop resistance. To improve the response rates and to consolidate in the time the duration of responses to IDH inhibitors, combination therapies are under exploration and are performed using a strategy based on the principle that these therapeutic associations could be capable to target leukemic clones and subclones driven both by IDH-dependent and IDH-independent mechanisms. In this context, two most likely associations are represented by IDH inhibitors with chemotherapy, hypomethylating agents, Bcl-2 inhibitors, and immune check inhibitors. The real and fundamental challenge of future studies will consist of demonstrating a significant improvement in survival and/or other long-term outcomes of MDS and AML patients treated with IDH inhibitors alone or in combination with other anti-leukemic drugs. This objective can be reached only through randomized phase III/IV clinical studies carefully conceived. It will be also fundamental to obtain data from ongoing studies about the proportion and the phenotypic and molecular characteristics of patients achieving complete remission without MRD, a finding that can predict a long-term benefit.

## Figures and Tables

**Figure 1 cancers-12-02427-f001:**
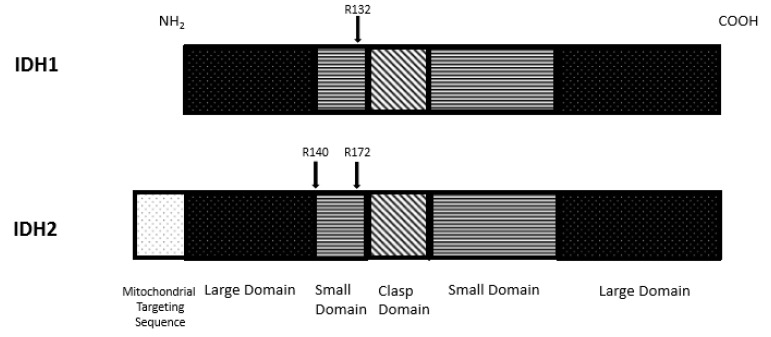
Domain map structure of isocitrate dehydrogenase (IDH) enzymes. IDH1 and IDH2 are composed of three different domains: Large domain, small domain, and clas domain. IDH2 contains also a 39 amino acid mitochondrial targeting sequence. The amino acids most frequently involved in *IDH* mutations are shown: arg 132 for IDH1; arg 140 and arg 172 for IDH2.

**Figure 2 cancers-12-02427-f002:**
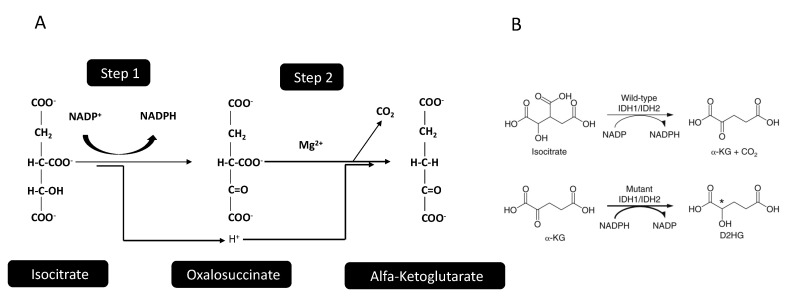
Enzymatic reactions catalyzed by wild-type and mutant IDH1 and IDH2 enzymes. (**A**) Normal IDH1 and IDH2 enzymes catalyze a two-step reaction. In the first step, isocitrate is oxidized to an unstable intermediate compound (oxalosuccinate), with concomitant reduction of NADP^+^ to NADPH. In the second step, the oxalosuccinate loses its beta-carbonyl group, which is released as CO_2_, giving rise to the formation of α-ketoglutarate (α-KG). The two H^+^ atoms generated during conversion of isocitrate to oxalosuccinate are used for NADP^+^ reduction to NADPH and for conversion of oxalosuccinate to α-KG. (**B**) Mutant IDH1 and IDH2 enzymes catalyze the reductive conversion of α-KG to (R)-2-hydroxyglutarate (D-2-HG) with concomitant oxidation of NADPH to NADP^+^. α-KG and D-2-HG are very similar from a structural point of view and differ only for the replacement of the ketone group present in α-KG, with the hydroxyl group present in D-2-HG.

**Figure 3 cancers-12-02427-f003:**
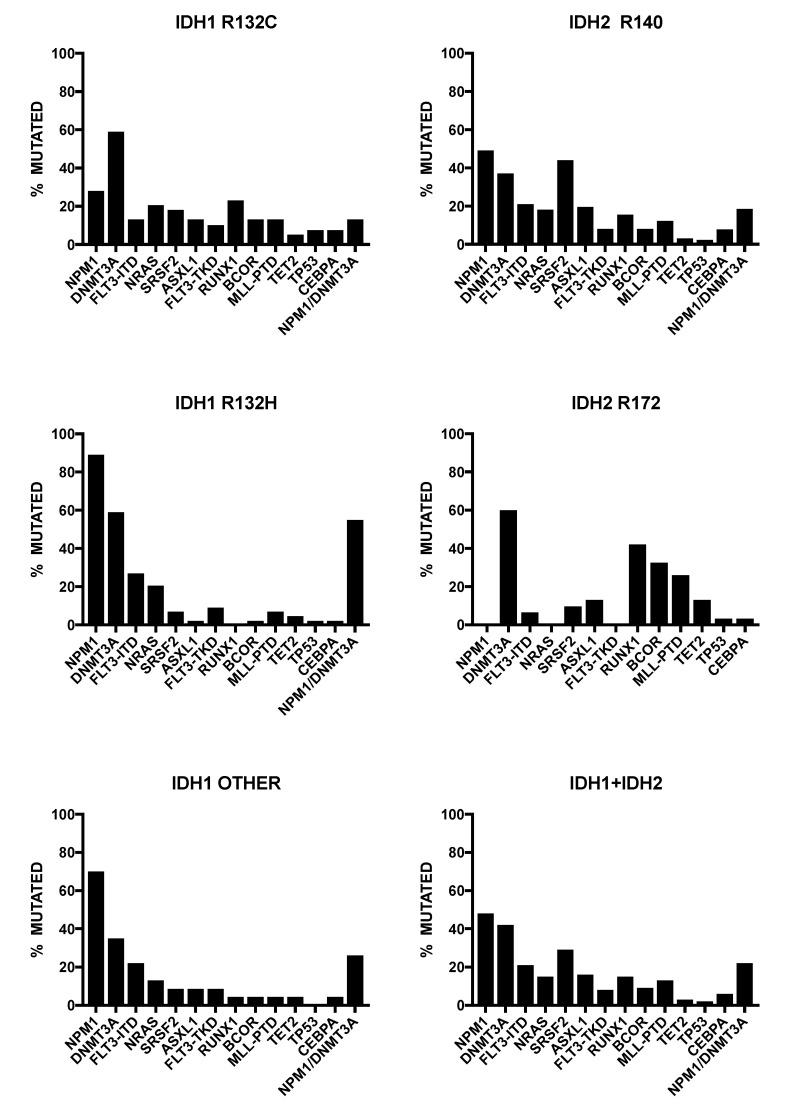
Co-mutations observed in IDH1 and IDH2-mutated acute myeloid leukemias (AMLs). The co-mutation pattern for the most frequently mutated genes in AMLs bearing IDH1-R132C, IDH1-R132H, or other IDH1 mutations is shown (data reported in Falini et al. [33]. The co-mutation pattern for the most frequently mutated genes in AMLs bearing IDH2 R140 or ID172 mutations and IDH1+IDH2 mutations is shown (data reported in Meggendorfer et al. [34]).

**Table 1 cancers-12-02427-t001:** IDH1 and IDH2 mutations in AML subtypes.

Reference	Number of Patients	AML Subtype	IDH Mutational Status
Mason et al. [40]	239	NPM1-mutated	Monocytic type (39%): IDH1 10.5%; IDH2 11%
			Non-monocytic type (30%): IDH1 14%; IDH2 14%
			CD34^−^/HLA-DR^−^ type (31%): IDH1 29.5%; IDH2 36%
Mason et al. [36]	84	NPM1-mutated	APL-like phenotype (47.5%) IDH1 30%; IDH2 30%
			Non APL-like phenotype (52.5%) IDH1 18%; IDH2 9%
Dunlap et al. [41]	20	NPM1-mutated	NPM1^+^/DNMT3A^+^/IDH1^+^ 4/20; NPM1^+^/IDH1^+^ 1/20
			NPM1^+^/DNMT3A^+^/IDH2^+^ 0/20; NPM1^+^/IDH2^+^ 3/20
Alpermann et al. [42]	660	NPM1-mutated	Type A NPM1 mutation (69%): IDH1 15%; IDH2 15%
			Type B NPM1 mutation (11%): IDH1 31%; IDH2 10%
			Type D NPM1 mutation (8%): IDH1 12%; IDH2 26%
Cocciardi et al. [43]	129	NPM1-mutated (paired at diagnosis and relapse)	Diagnosis: IDH1 22.5%; IDH2 18.6% (86% concordance)
			Relapse: IDH1 22.5%; IDH2 17% (88% concordance)
Sun et al. [44]	80	MLL-PTD	IDH1 11%; IDH2 20%
Al Hinai et al. [45]	85	MLL-PTD	IDH1 18.8%; IDH2 21.2%
Gaidizik et al. [46]	140	RUNX1-mutated	IDH1 9.5%; IDH2 17.5%
Haferlach et al. [47]	152	RUNX1-mutated	IDH1 8.5%; IDH2 15.8%
Eisfeld et al. [48]	23	Trisomy 11	IDAH1 9%; IDH2 39%

**Table 2 cancers-12-02427-t002:** Ongoing clinical trials of IDH2 inhibitors in AML and myelodysplastic syndromes (MDS).

Trial Identification (Sponsor)	Clinical Phase	Title	Disease and Objectives	Drugs	Status
NCT 01915498 (Agios)	Phase I, Phase II	Phase 1/2 study of AG-221 in subjects with advanced hematologic malignancies with an IDH2 mutation	Advanced AML Safety, tolerability, MTD	AG-221 (Enasidenib)	Active, not recruiting
NCT 02577406 (Celgene)	Phase II	An efficacy and safety study of AG-221 (CC-90007) versus conventional care regimen in older subjects with late stage AML harboring an IDH2 mutation (IDHENTITY)	AML ≥60 years OS, ORR, EFS, duration of response, time to response	Enasidenib, BSC, azacitidine, low-dose AraC, intermediate-dose AraC	Active, not recruiting
NCT 02632708 (Agios, Celgene)	Phase I	Safety study of AG-120 or AG-221 in combination with Induction and consolidation therapy in participants With newly diagnosed acute myeloid leukemia (AML) With an IDH1 and/or IDH2 mutation	Newly diagnosed AML, AML arising from MDS, antecendent hematologic disorder or therapy Safety, tolerability, MTD	Ivosidenib or enasidenib plus standard chemotherapy	Active, not recruiting
NCT 02677922 (Celgene)	Phase I/Phase II	A safety and efficacy of oral AG-120 plus subcutaneous azacitidine and oral AG-221 plus subcutaneous azacitidine in subjects with newly diagnosed AML	AML DLT, Safety, Pharmacokinetics	Ivosidenib or enasidenib plus azacytidine	Active, not recruiting
NCT 03383575 (MD Anderson Cancer Center)	Phase II	Azacitidine and enasidenib in treating patients with IDH2-mutant myelodysplastic syndrome	High-risk MDS, R/R MDS Safety, ORR, EFS, OS	Enasidenib plus azacitidine (arm 1, HNA-naive MDS), enosidenib (arm 2, R/R MDS)	Active, recruiting
NCT 03515512 (Massachusetts General Hospital)	Phase I	IDH2 inhibition using enasidenib as maintenance therapy for IDH2-mutant myeloid neoplasms following allogeneic stem cell transplantation	IDH2 mutant myeloid neoplasms	Enasidenib	Active, recruiting
NCT 03683433 (MD Anderson Cancer Center)	Phase II	Enasidenib and zacitidine in treating patients with recurrent or refractory AML and IDH2 gene mutation	R/R AML ORR, EFS, OS	Enasidenib plus azacitidine	Active, recruiting
NCT03728335 (City of Hope Medical Center)	Phase I	Enasidenib as maintenance therapy in treating patients with AML with IDH2 mutation after donor stem cell transplant	AML in post HCT Safety, tolerability, OS, EFS	Enasidenib	Active, recruiting
NCT 03744390 (Groupe Francophone des Myelodysplasies)	Phase II	IDH2 (AG221) inhibitor in patients with IDH2 mutated myelodysplastic syndrome	High-risk, R/R MDS ORR, Duration of response	Enasidenib	Active, recruiting
NCT 03825796 (Jonsson Comprehensive Cancer Center)	Phase II	CPX-351 and enasidenib in treating patients with released AML characterized by IDH2 mutation	Relapsed AML Remission rate, hematological toxicity	Enasidenib plus CPX-351 (liposome-encapsulated daunorubicin-cytarabine)	Active, recruiting
NCT 03839771 (Stichting Hemato-Oncologie voor Wolkvassenen, NL	Phase III	A study of ivosidenib or enasidenib in combination with induction therapy and consolidation therapy, followed by maintenance therapy in patients with newly diagnosed AML or myelodysplastic syndrome EB2, with an IND1 or IDH2 mutation, respectively, eligible for intensive chemotherapy (HOVON 150 AML)	IDH1 or IDH2 mutant AML or MDS EFS, OS, RFS	Ivosidenib or enasidenib plus standard chemotherapy	Active, recruiting
NCT 03881735 (Roswell Park Cancer Institute)	Phase II	Enasidenib in treating patients with relapsed or refractory AML with an IDH2 gene mutation	RR/AML EFS, OS	Enasidenib after salvage chemotherapy	Active, recruiting
NCT 04203316 (Children’s Oncology Group)	Phase II	Enasidenib for the treatment of relapsed or refractory AML patients with an IDH2 mutation	Pediatric AML Safety, tolerability, pharmacokinetics	Enasidenib	Active, recruiting

Abbreviations: EFS: Event-free survival; OS: Overall survival; ORR: Overall response rate; CRR: Complete response rate; DLT: Dose limiting toxicity; MTD: Maximum tolerated dose; RFS: Relapse-free survival; PFS: Progression-free survival.

**Table 3 cancers-12-02427-t003:** Ongoing clinical trials of IDH1 inhibitors in AML and MDS.

Trial Identification (Sponsor)	Clinical Phase	Title	Disease and Objectives	Drugs	Status
NCT 02074839 (Agios)	Phase I	Study of orally administered AG-120 in subjects With advanced hematologic malignancies with an IDH1 mutation	R/R AML, untreated AML, MDS, other IDH1-mutated hematologic malignancies Safety. Tolerability, MTD	Ivosidenib	Active, recruiting
NCT 02632708 (Agios, Celgene)	Phase I	Safety study of AG-120 or AG-221 in combination with induction and consolidation therapy in participants With newly diagnosed acute myeloid leukemia (AML) With an IDH1 and/or IDH2 mutation	Newly diagnosed AML, AML arising from MDS, antecendent hematologic disorder or therapy Safety, tolerability, MTD	Ivosidenib or enasidenib plus standard chemotherapy	Active, not recruiting
NCT 02677922 (Celgene)	Phase I/Phase II	A safety and efficacy of oral AG-120 plus subcutaneous azacitidine and oral AG-221 plus subcutaneous azacitidine in subjects with newly diagnosed AML	AML DLT, Safety, Pharmacokinetics	Ivosidenib or enasidenib plus azacitidine	Active, not recruiting
NCT 03471260 (MD Anderson Cancer Center)	Phase I/Phase II	Ivosidenib and venetoclax with or without azacitidine in treating participants with IDH1 mutated hematologic malignancies	R/R AML, high-risk MDS, myeloproliferative neoplasms Safey, MTD, Pharmacokinetics, ORR, EFS, OS	Ivosidenib plus venetoclax with or without azacitidine	Active, recruiting
NCT 03564821 (Massachusetts General Hospital, Agios)	Phase I	IDH1 inhibition using ivosidenib as maintenance therapy for IDH1-mutant myeloid neoplasms following allogeneic stem cell transplantation	IDH1 mutant myeloid neoplasms MRD, safety, GVHD, IDH clonal evolution	Ivosidenib	Active, recruiting
NCT 03839771 (Stichting Hemato-Oncologie voor Wolkvassenen, NL	Phase III	A study of ivosidenib or enasidenib in combination with induction therpy and consolidation therapy, followed by maintenance therapy in patients with newly diagnosed AML or myelodysplastic syndrome EB2, with an IND1 or IDH2 mutation, respectively, eligible for intensive chemotherapy (HOVON 150 AML)	IDH1 or IDH2 mutant AML or MDS EFS, OS, RFS	Ivosidenib or enasidenib plus standard chemotherapy	Active, recruiting
NCT 04056910	Phase II	A study of the IDH1 inhibitor AG-120 in combination with the checkpoint blockade inhibitor, Nivolumab, in patients with IDH1 mutated relapsed/refractory AML and high-risk MDS	R/R AML, high-risk MDS DLT, Best OR, PFS	Ivosidenib plus nivolumab	Active, recruiting
NCT 176393 (CStone Pharmaceuticals)	Phase I	A China bridging study of ivosidenib in R/R AML subjects with an IDH1 mutation	R/R AML Pharmacokinetics, ORR	Ivosidenib	Active, recruiting
NCT 04250051 (Northwestern University, Agios)	Phase I	Ivosidenib and combination chemotherapy for the treatment of IDH1 mutant relapsed or refractory AML	RR/AML, R/R MDS MTD, safety, CRR	Ivosidenib plus standard chemotherapy	Active, recruiting
NCT 02719574 (Forma Therapeutics)	Phase I, Phase II	Open-label study of FT-2102 with or without azacitidine in patients with AML or MDS with an IDH1 mutation	AML, MDS MTD, DLT, CRR	FT-2102 (olutsidenib) plus azacitidine or cytarabine	Active, recruiting
NCT 04013880 (Vanderbilt-Ingram Cancer Center, Astex and Forma Therapeutics)	Phase I/Phase II	ASTX 727 and FT-2102 in treating IDH1-mutated recurrent/refractory myelodysplastic syndrome or AML	R/R AML, R/R MDS Safety, ORR	FT-2102 (olutasidenib) plus ASTX 727 (DNMT inhibitor)	Active, recruiting
NCT 03127735	Phase I	BAY 1436032 in patients with mutated IDH1(mIDH1) advanced acute myeloid leukemia (AML)	Advanced AML MTD, Objective efficacy response	BAY 1436032 Safety and tolerability,	Active, not recruiting

Abbreviations: EFS: Event-free survival; OS: Overall survival; ORR: Overall response rate; CRR: Complete response rate; DLT: Dose limiting toxicity; MTD: Maximum tolerated dose; RFS: Relapse-free survival; PFS: Progression-free survival.

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
