# Peer review of "Isocitrate Dehydrogenase Mutations in Myelodysplastic Syndromes and in Acute Myeloid Leukemias"

_cancers, 2020, doi:10.3390/cancers12092427_

Round 1

Reviewer 1 Report

Manuscript is improved, but still difficult to read- It is text- heavy, and would benefit with synthesis of some data into tabular format (1). It also remains unnecessarily long and with large sections that do not correlate to the topic at hand (2). It also needs to be edited for English/ grammar (significant).

1) Ex for a table: when talking about co-mutation status. As written, it is difficult to read, but if it is organized with comutation frequency/ any known prognostic data/ reference ( or something like that) you would cut down on words and make it more reader friendly.

2) Still packed with uneccessary/ off topic data: I am providing an example below: how is anything in the paragraph below relevant to the topic of IDHm malignancies? This is just one example of many.

Venetoclax-based therapy can induce significant responses in about 70% of patients; however, primary and secondary mechanisms of resistance occur during venetoclax treatment, thus indicating that an understanding of these mechanisms is required. A recent study by Pei et al., provided evidence that monocytic AMLs (FAB M5) are resistant to venetoclax plus azacytidine treatment compared non-monocytic AMLs (62% vs 8%, respectively of drug resistance); in vitro studies showed that leukemic cell populations enriched in stem cells (ROS-low/CD34+) are resistant to venetoclax + azacitidine [197]. Importantly, monocytic AMLs displayed a low expression of Bcl-2, thus explaining the low sensitivity ofthese leukemias to venetoclax [197 -

I can say I am very enthusiastic about this topic, and I believe you cover a lot of content, but the lack of overall direction and readability made it almost impossible to read in 1 sitting.

I commend the authors on collection of a lot of relevant data that could be considered an update on prior reviews but the presentation of this data still needs work.

Author Response

We have made an effort to simplify the text, with particular emphasis to the sections related to IDH mutations in AML and BCL-2 targeting as indicated by this reviewer.

IDH mutations in AML – The length of this section was considerably shortened. Now all the analysis related to the pattern of IDH1 and IDH2 mutations in various AML subtypes was included in a new Table III and the relative text was just reduced to a sentence. Furthermore, to better clarify the complex topic of IDH1/IDH2-comutations a new figure (Fig.3) was now introduced. We hope that all these changes, in line with the reviewer’s suggestions, may definitely help to simplify the section on IDH mutations in AML.

Bcl-2 targeting – This section was considerably shortened. The paragraph on monocytic AMLs in the resistance to BCL-2 inhibitors was completely deleted. General results on the therapeutic effect of Venetoclax on AMLs were also deleted. Thus, now this section is strictly focused on the analysis of the effects of Venetoclax on IDH-mutated AMLs.

Furthermore, the length of the manuscript was reduced from 55 to 46 pages. All the sections of the manuscript were shortened and strictly focused to IDH.

The English was revised.

Reviewer 2 Report

Authors addressed my concerns.

Just one minor comment for this updated version. The mosaic style to show protein domain in figure 1 could be instead of pfam style (e.g., hhttps://pfam.xfam.org/family/IDH#tabview=tab1).

Author Response

OK

Reviewer 3 Report

The legend for figure 1 is missing but instead the figure 2 legend has been used

Author Response

Table I legend was modified and replaced with the correct figure legend.

This manuscript is a resubmission of an earlier submission. The following is a list of the peer review reports and author responses from that submission.

Round 1

Reviewer 1 Report

Review Summary

The review entitled “ Isocitrate dehydrogenase mutations in myelodysplastic syndromes and in acute myeloid leukemias” aims to summarize the available data on the role of IDH 1 and 2 mutations in myeloid malignancies.

Broad Comments

  • While quite comprehensive, this review lacks focus and is difficult to read. The review is quite disorganized, and while it contains a lot of the relevant information, would not be my go to source to find any of it due to organization.
  • Each section has a laundry list of papers- nothing is tied together. A key feature of a good review is to have a summation or guide the audience of how to understand the literature. This review seems like a list of various studies without synthesis. Even the conclusion is a list of ongoing new trials, not a summation.
  • There is a lot of jumping from topic to topic, as well as meandering around a topic with information that is completely out of the scope of the topic.
  • I cannot identify a good target audience- are you reaching out to clinicians? Lab researchers? Medical students and trainees? It does not satisfy any demographic as currently written.
  • Why 42 pages? Much of what is written is not even about the topic. Cut sections and read- remove things that are not about IDH and do not strengthen your point about IDH.

Specific comments

  • Unfortunately this entire manuscript is poorly written and I cannot include all examples (it would require re-writing all 42 pages . Here are a few ( please note that the feedback is repetitive themes in all sections so spelled out in the first few, and then more point errors noted..
  • Paragraph 1 ( lines 27-37): Starts with IDH genes in sentence 1- jumps to protein localization in sentence 1. Functioning state in sentence 3. Enzyme mechanism of action thereafter. After mechanism we jump back to molecular structure of each enzyme in line 77- why? The intro also does not refer at all to the title of the paper and why they are reviewing IDH in AML and MDS.
  • Repetition: Line 31 on- localization of enzymes. Line 117: again localization of enzyme. Why? The purpose is completely unclear
  • The introduction is (and should be) more dedicated to established mechanism- suddenly in line 74 we are talking about pt derived chondroma cell experiments- that really is insufficient data unless it is presented as “postulated” etc.
  • Line 82: separate domainsof (spacing error)
  • Line 122 is the title: IDH in AML- this is followed by No text. Line 123 Straight is subheading IDH mutations in clonal hematopoeisis. While arguably this is related, it is its own beast. Almost the whole section ( barring maybe 2 sentences) has nothing to do with the role of IDH mutations in this scenario, but is a summary of some data in ARCH ( or CHIP)
  • Line 192-224: again, nothing to do with the topic. I am a leukemia physician- I do not require an entire paragraph about MDS classification. If this article is guided toward a non-med student audience, this is inappropriate. It is also not reader friendly for the lab researcher, as it is devoid of informative details.
  • Line 224: the author has not introduced the overall incidence of IDH mutations yet (this comes in line 237 ) but immediately jumps into IDH and trisomy 8- why? It is unclear. The entire rest of the section is study after study without synthesis of data. It is difficult to synthesize molecular data without pathogenic rationale (which they could have included hypotheses but didn’t) OR simply made a table of associations.
  • Line 292: immature progenitor defined as (MP). Then used in the next sentence as IMP. Inconsistent.
  • IF the author chooses to re-write this review per the title- I would suggest removing all the unrelated topics. Within IDH and AML have subsections ex: IDH and associated cytogenetic abnormalities, IDH and associated mutations, etc.
  • Line 631: This is called AML with myelodysplasia related changes. Same comments from prior sections.
  • Line 765: spelling short.
  • Line 772- watch the abbreviations/ make sure to define all. Ex: second MN
  • Line 941: percentages error.
  • Section of IDH and DNA methylation- Again- consider restructuring; you were in the middle of describing IDH from a clinical context, and there was a sudden switch ( without any context) to epigenetic implications? Then in line 1001 you jump back to prognosis? Then 1068 is leukemogenesis- Structure either scientifically or clinically. Leukemogenesisà MDSà AML typesà prognosticà value of MRDà etc.
  • 1256: you jump to IDH inhibitors without addressing a switch to therapeutic strategy- given the rest of your paper, this you should probably explain allosterics here, since it directly relates to mechanisms of resistance. This resistance section is also incomplete.
  • Repeating the same theme- lot of unnecessary off topic information. Ex: M5 disease may be less responsive to ven/ Aza ( line 1596 on) This adds nothing- you allude a little to potential reasons it could relate which is valid, but this should be 1 sentence, not 2/3 of a page!
  • Line 1700- I believe you mean Olaparib ( it says enasidenib)

This was outside the scope at submission, but include recently published info: https://ashpublications.org/bloodadvances/article/4/9/1894/454778/Molecular-mechanisms-mediating-relapse-following

Author Response

1 – Introduction. The reviewer has raised a number of serious criticisms concerning the whole organization of the Introduction section. Particularly, most of criticisms were related to the lack of organization of this section. To meet this reasonable and acceptable criticism, all the introduction section was completely reorganized and presented according to a logical sequence, starting from genes, to molecular structure of IDH enzymes, to enzymatic activities and cellular localization of IDH enzymes. Furthermore, the refence to chondroma study in line 74 and the relative sentence was removed. Finally, the repetitive sentence in line 117 on enzyme localization was removed.

2 – IDH mutations in clonal hematopoiesis. This section was shortened; however, we believe that this section is relevant in the context of the analysis of IDH mutations in myeloid neoplasia.

3 – IDH mutations in myelodysplastic syndromes. In line with a specific suggestion of this reviewer, all the paragraph concerning MDS classification was removed. The whole section was shortened. At the end of this section, a sentence was added indicating the main pathogenic role played by IDH1/IDH2 mutations in MDS.

4 – Reorganization of the sequence of the sections leukemogenesis, methylation, prognosis and MRD. As suggested by this reviewer, these four sections were reorganized according to a more logical sequence following this order: Leukemogenesis ® Methylation ® Prognosis ® MRD.

5 - IDH inhibitors – This section was now introduced with two sentences supporting the therapeutic strategy based on IDH inhibitors. Now in the description of IDH1 and IDH2 inhibitors is explained the mechanism of allosteric inhibition, this mechanism being relevant for both understanding how these inhibitors block the activity of mutant enzymes and to understand the mechanisms of resistance.

6 - Ventoclax/Azacitidine – The low sensitivity of M5 AMLs to Ven/Aza was now just mentioned.

7 - Minor and specific points – All have been modified according to the suggestions of the reviewer.

Reviewer 2 Report

In the paper entitled “isocitrate dehydrogenase mutations in myelodysplastic syndromes and in acute myeloid leukemias”, authors summarized IDH mutations and related inhibitors in AML. This review did a very good job in summarizing published studies. This reviewer has only one major comment: what is the outstanding questions and challenge for IDH mutation in AML treatment? Discuss this would largely increase impact of this review and aid advances of this filed.

Author Response

To better define the real and fundamental challenge related to the study of IDH mutations and inhibitors in leukemkia, the following sentence was added at the end of the conclusions:

The real and fundamental challenge of future studies will consist to demonstrate a significant improvement in survival and/or other long-term outcomes of MDS and AML patients treated with IDH inhibitors alone or in combination with other anti-leukemic drugs. This objective can be reached only through randomized phase III/IV clinical studies carefully conceived. It will be also fundamental to obtain data from ongoing studies about the proportion and the phenotypic and molecular characteristics of patients achieving complete remission without MRD, a finding that can predict a long-term benefit.

Reviewer 3 Report

The review is very comprehensive, however the different parts are way too long and thus difficult to read. I would suggest to cut down at least 2/3 and to limit the review to important observations/publications rather than reporting every single paper written on the respective topic. For example, the paragraphs explaining the IDH mutations can be summarized in tables. Furthermore, while the paragraphs explaining the inhibitors are interesting they give too much information, esp.since this is not the main topic of the review these parts should be shortened drastically. 

General comments:

  • not enough references, especially in the first paragraphs. This needs to be revised thoroughly since referencing is one of the most important parts of reviews
  • if the authors consider that the part with the structure should remain in the shortened manuscript, an image of the structure of IDH1 and IDH2 might be useful
  • revise that all gene names are written in italic

Specific comments:

  • paragraph starting at line 108 seems to have been written by another author since it introduces IDH again... Thus, linking of different parts need to be improved and repetition avoided.
  • in line 146 the term CHIP appears for the first time without defining it beforehand
  • line 158 - 161: only state official gene names thus abbreviations, not the full names
  • avoid using ":" or ";" but rather make independent sentences, otherwise the sentences get too long and it is very difficult to read

Author Response

The size of the manuscript was now reduced of 10 pages. We prefer to leave in the manuscript a detailed description of the various AML subtypes exhibiting IDH mutations because this is a very important topic not only at the level of understanding of leukemia heterogeneity, but also for the important implications at the level of targeted therapy.

The whole introduction section was now reorganized according to the comments of the other review and now structured according to a logical sequence: starting from genes, to molecular structure of IDH enzymes, to enzymatic activities and cellular localization of IDH enzymes. Furthermore, according to a specific request of this reviewer, a figure with the structure of IDH1 and IDH2 was now included.

Reviewer 4 Report

The study is too extensive and overwelming, and due to lack of structure, figures and tables, the manuscript can not be used to look up specific details about IHD. 

Discussing all recurrent mutations in MDS and AML in context of IDH mutations is very ambitious. It is difficult to see the possible benefit for the reader as the manuscript is formed. 

Combining MDS and AML is also a challenge, since both MDS and AML are heterogenous and not necessarily related in all subsets. 

The abstract state that «novel therapies based on the specific targeting of mutant IDH have revolutioned the therapy of these patients». This is is false to this date, since the randomized phase clinical III trials hardly have started recruiting patients». 

Some of the references on cytogenetics and risk classes seems limited or wrong. The text has limited quality of language and include frequent misspelling. 

This project need focus and restructuring to help the reader. 

Author Response

  • The manuscript was now restructured in a more logical sequence, thus helping the reader to more easily understand this complex topic.
  • We do not agree that the discussion of all the recurrent mutations in MDS and AML in the context of IDH mutations is too ambitious. This is simply an honest attempt to provide available information on the AML subsets, following the principle that a better understanding of leukemia heterogeneity is a fundamental tool to try to define new therapeutic approaches.
  • We agree with this criticism and therefore we have now modified the sentence in the abstract, stating that “novel therapies based on targeting of mutant IDH may contribute to the development of more efficacious treatments of these patients”.
  • Risk classes of AML patients are now analyzed and misspelling errors have been corrected.